# OCN: Effectively Utilizing Higher-Order Common Neighbors for Better Link Prediction

**Juntong Wang**[1,2]   **Xiyuan Wang**[1,2]   **Muhan Zhang**[1]*
[1]Institute for Artificial Intelligence, Peking University
[2]School of Intelligence Science and Technology, Peking University
jtwang25@stu.pku.edu.cn, {wangxiyuan,muhan}@pku.edu.cn

## Abstract

Common Neighbors (CNs) and their higher-order variants are important pairwise features widely used in state-of-the-art link prediction methods. However, existing methods often struggle with the repetition across different orders of CNs and fail to fully leverage their potential. We identify that these limitations stem from two key issues: redundancy and over-smoothing in high-order common neighbors. To address these challenges, we design orthogonalization to eliminate redundancy between different-order CNs and normalization to mitigate over-smoothing. By combining these two techniques, we propose Orthogonal Common Neighbor (OCN), a novel approach that significantly outperforms the strongest baselines by an average of 7.7% on popular link prediction benchmarks. A thorough theoretical analysis is provided to support our method. Ablation studies also verify the effectiveness of our orthogonalization and normalization techniques. Code is available at: ⬤ https://github.com/qingpingmo/OCN

## 1 Introduction

The application of link prediction spans numerous fields. For example, it can be used to forecast website hyperlinks [Zhu et al., 2002]. In bioinformatics, it plays a critical role in analyzing protein-protein interactions (PPIs) [Airoldi et al., 2008]. Similarly, in e-commerce, link prediction is a core component in developing recommendation systems [Huang et al., 2005, Lü et al., 2012]. Currently, the most popular link prediction models are based on Graph Neural Networks (GNNs). The first GNN for link prediction was the Graph AutoEncoder (GAE) [Kipf and Welling, 2016]. It uses the inner product of the two target nodes' representations, produced by a Message Passing Neural Network (MPNN) [Gilmer et al., 2017], as the logits for the probability that a link exists between the two nodes. Despite its success on some citation graphs, such as Cora [Sen et al., 2008], GAE computes the representations of two nodes separately, thus failing to capture structural relationships between them, such as the number of common neighbors (i.e., nodes connected to both target nodes by an edge), which is a crucial heuristic in link prediction.

To address the inability to capture pairwise structural relationships, various methods have been proposed [Zhang et al., 2021, Yun et al., 2021, Chamberlain et al., 2023, Wang et al., 2024]. Although these methods differ in their detailed implementations, they all focus on computing the neighborhood overlap between the two target nodes, which includes both common neighbors and **higher-order common neighbors** (nodes connected to the two target nodes via a walk or path). While higher-order common neighbors provide auxiliary information to common neighbors and improve performance in some cases [Yun et al., 2021, Chamberlain et al., 2023], they have not yet been widely adopted and do not always improve performance due to two key problems:

---

*Correspondence to Muhan Zhang

39th Conference on Neural Information Processing Systems (NeurIPS 2025).

The first problem is **redundancy**: different-order common neighbors of the same node pair may overlap significantly. A node can be a common neighbor and a higher-order common neighbor of some node pair at the same time, as the path or walk connecting them may not be unique. This overlap makes higher-order CNs less informative when common neighbors are already used.

The second problem is **over-smoothing**. In the context of node classification problems [Oono and Suzuki, 2020], over-smoothing describes the phenomenon that, as the number of GNN layers increases, all nodes have similar representations because their neighborhoods become more and more similar. Here, over-smoothing means that, as the order of common neighbors increases, a node can become a high-order CN for more and more node pairs simultaneously. When the path/walk length is sufficiently large, the high-order common neighbors of a node pair will encompass the entire graph. At this time, aggregating the features/embeddings of the high-order common neighbors makes every node pair have similar pairwise representations, leading to over-smoothing in the context of pairwise representation learning.

Both issues hinder the effective utilization of higher-order common neighbors, thereby limiting the learning of complex pairwise structures and preventing state-of-the-art link prediction models from achieving optimal performance. For example, Wang et al. [2024] found that utilizing only first-order common neighbors led to the best performance in their models. To address these two problems, we propose two techniques: **coefficient orthogonalization** and **path-based normalization**, respectively.

For **coefficient orthogonalization**, which solves redundancy, we remove the linear correlation between the coefficients of different-order common neighbors. For instance, given a pair of nodes in a graph with $n$ nodes, the coefficient indicating whether each node participates in some order of common neighbors of this node pair becomes a vector $\in \mathbb{R}^n$. We use the *Gram-Schmidt orthogonalization* process to *eliminate the correlation between the coefficient vectors of different-order common neighbors*, so that models can better leverage information from higher-order CNs. The coefficients can be binary (0 or 1) to indicate whether a node is a common neighbor, or functions of node degrees as used in [Yun et al., 2021], or the number of walks of a certain length connecting the two target nodes in which the node participates, as used in this work. With coefficient orthogonalization, our model significantly outperforms previous link prediction models using common neighbors. Furthermore, to accelerate the orthogonalization process, we propose a *polynomial trick*, which achieves similar performance to precise orthogonalization while eliminating the extra computational overhead.

For **path-based normalization**, which addresses over-smoothing, we *divide the coefficient of each node by the number of $k$-hop walks* in which it participates. The intuition behind the normalization is that when a node participates in a large number of walks, it will more frequently appear in the common neighbors of other node pairs, which causes the features of different links to become similar, leading to over-smoothing. Notably, when the path or walk has a length of 1, the path count reduces to the node degree, and our normalized CN *degenerates to a famous link prediction heuristic, Resource Allocation* [Zhou et al., 2009]. To theoretically analyze path-based normalization, we use a random graph model and prove that, with the normalization, $k$-hop CNs lead to a strictly decreasing upper bound of the proximity of positive node pairs (real links) with the increasing of $k$, while without the normalization, high-order CNs do not decrease the proximity no matter how large $k$ is used. This result justifies the effectiveness of using **normalized** high-order CNs, and potentially explains why previous methods using high-order CNs do not always work (due to lack of normalization). In practice, we use a method similar to batch normalization [Ioffe and Szegedy, 2015] to estimate the number of walks efficiently, avoiding the computational overhead of exact counting.

By combining both techniques with previous methods [Wang et al., 2024], we propose **Orthogonal Common Neighbor** (**OCN**) and its variant with an approximated and faster orthogonalization process, Orthogonal Common Neighbor with Polynomial Filters (OCNP). In our experiments, the performance of OCN and OCNP significantly outperforms existing models, achieving state-of-the-art results on several Open Graph Benchmark datasets [Hu et al., 2020]. Ablation studies also verify the effectiveness of orthogonalization and normalization.

## 2 Preliminaries

For an undirected graph $G = (V, E, A, X)$, where $V = \{1, 2, \ldots, n\}$ represents the set of $n$ nodes, $E \subseteq V \times V$ denotes the edge set, $X \in \mathbb{R}^{n \times F}$ is a matrix of node features, and $A \in \mathbb{R}^{n \times n}$ is the symmetric adjacency matrix. The entries of the adjacency matrix are defined such that $A_{uv} = 1$

if there is an edge $(u, v) \in E$, and $A_{uv} = 0$ otherwise. This adjacency matrix captures the direct connections between nodes in the graph. The degree of a node $u$, denoted by $d(u, A)$, is defined as the sum of the entries in the corresponding row of the adjacency matrix, i.e., $d(u, A) := \sum_{v=1}^{n} A_{uv}$, which represents the number of neighbors of node $u$.

We further define $A^l$ as the higher-order adjacency matrix, where the entry $A_{uv}^l$ represents the number of walks of length $l$ between nodes $u$ and $v$. Specifically, $A^l$ encapsulates more complex relationships between nodes, extending beyond direct neighbors to capture connections that involve intermediary nodes. The matrix $A^l$ can be computed by raising the adjacency matrix $A$ to the power $l$, where higher powers encode longer walks between nodes.

The $k$-order neighbor set $N(u, A^l)$ is defined as the set of all nodes that are reachable from node $u$ through a walk of length $l$, i.e., $N(u, A^l) = \{v \mid v \in V, A_{uv}^l > 0\}$.

This set includes all nodes that can be reached from $u$ by traversing $l$-length walks, thereby expanding the notion of proximity beyond direct neighbors. While some methods define high-order neighbors based on paths or shortest paths, we adopt a different approach. Specifically, we do not rely on shortest paths.

In the context of $k$-hop neighbors, the $k$-hop common neighbors of two nodes $u$ and $v$, denoted as $\mathrm{CN}_k(u, v)$, are defined as:

$$\mathrm{CN}_k(u, v) = \bigcup_{\substack{2(k-1) < k_1 + k_2 \leq 2k, \\ k_1, k_2 \leq k}} \left( N_{k_1}(u) \cap N_{k_2}(v) \right), \tag{1}$$

where the union is taken over all pairs $k_1$ and $k_2$ such that $2(k-1) < k_1 + k_2 \leq 2k$. Here, $N_{k_1}(u)$ and $N_{k_2}(v)$ denote the sets of nodes reachable from $u$ and $v$ via walks of lengths $k_1$ and $k_2$, respectively. The concept of common neighbors is critical in graph-based models, as it quantifies the overlap between the neighborhood structures of two nodes, which is a useful measure for tasks such as link prediction.

## 3 Related Work

**Link Prediction Model**    Link prediction generally employs three main approaches: *Node embeddings*, *Link prediction heuristics*, and *GNNs*. *Node embeddings* aim to find an embedding for nodes such that similar nodes (in the original network) have similar embeddings [Perozzi et al., 2014, Belkin and Niyogi, 2001, Grover and Leskovec, 2016, Kazemi and Poole, 2018]. *Link prediction heuristics* [Newman, 2001, Barabâsi et al., 2002, Adar and Adamic, 2003, Zhou et al., 2009] mainly rely on handcrafted structural features for prediction. In recent years, methods based on *GNNs* have become a research hotspot. SEAL [Zhang and Chen, 2018] calculates the shortest path distance between nodes $i$ and $j$, extracts the $k$-hop subgraph, generates augmented features $X'$, applies MPNN to aggregate node representations, and predicts the link.

**Architecture Combining MPNN and SF**    The SF-then-MPNN framework, exemplified by SEAL [Zhang and Chen, 2018], enriches the input graph with structural features, which are then passed to MPNN to enhance expressivity. However, this approach requires re-running the MPNN for each target link, leading to lower scalability. In contrast, models such as NeoGNN [Yun et al., 2021] and BUDDY [Chamberlain et al., 2023] adopt the SF-and-MPNN framework, where the MPNN takes the original graph as input and runs only once for all target links, thus enhancing scalability. However, this approach sacrifices expressivity, as the structural features are detached from the final node representations. To address this, the MPNN-then-SF framework proposed by NCN [Wang et al., 2024] significantly improves performance by first running MPNN on the original graph and then employing structural features to guide the pooling of MPNN features, resulting in better expressivity while retaining high scalability.

## 4 Orthogonalization

We observe substantial redundancy between CNs of different orders. By analyzing the correlation coefficients between CNs at various orders, we find that this correlation increases

with the order, reaching high levels, as shown in Figure 1. This indicates that the current definition of CNs contains significant linear dependencies across different orders. In contrast, higher-order CNs based on shortest path distances (SPDs) are inherently independent, as a node cannot simultaneously belong to the common neighbor sets of different SPDs.

The presence of such redundancy negatively impacts model performance by reducing the model's expressive power. When different orders of CNs become highly correlated, it becomes difficult for the model to effectively differentiate between them, limiting its ability to capture distinct structural relationships. This not only impedes the model's capacity to learn from richer, higher-order interactions, but also diminishes its generalization ability, preventing it from uncovering subtle but important relationships in the graph.

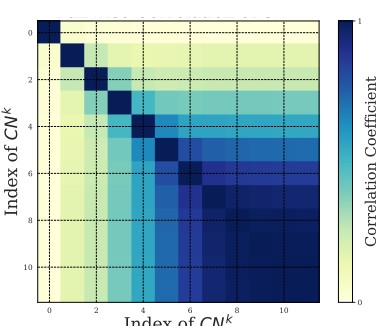

Figure 1: The heatmap illustrates the correlation coefficients between different orders of $k$-order Common Neighbors (CNs), highlighting increasing redundancy with higher orders.

## 4.1 Scalable Orthogonalization

To mitigate the negative effects of redundancy, we propose using the Gram-Schmidt process to transform $k$-hop CNs into mutually independent representations. This approach maximizes the information content of each $CN^k$ matrix by eliminating linear correlations between different orders of CNs.

For $CN^k$, we define it based on Equation (1) as follows: $CN^k(u, v) = \sum_{\substack{2(k-1) < k_1 + k_2 \leq 2k, \\ k_1 \leq k, \ k_2 \leq k}} \left(A^{k_1}\right)_u \odot \left(A^{k_2}\right)_v$, where $CN^k(u, v) \in \mathbb{R}^n$, with its $i$-th element representing the number of $2k$- and $2k-1$-length walks from node $u$ to node $v$ that node $i$ participates in as an intermediate point. The result of orthogonalizing $CN^k(u, v)$ for $k = 1, 2, \ldots$ is referred to as $OCN^k$. The matrix $OCN^k$ can be interpreted as a set of orthogonalized representations for the $k$-hop CNs. The detailed definition and explanation of the correlation matrix can be found in Appendix B.

However, orthogonalizing $CN^k$ over the entire graph poses significant computational challenges. To address this, we draw inspiration from Batch Normalization [Ioffe and Szegedy, 2015], which maintains running estimates of batch statistics (mean and variance) to normalize activations during training. Similarly, we propose a strategy for orthogonalizing $CN^k$ across mini-batches. The core idea is to maintain a running inner product: $\hat{\xi}_t^i \leftarrow (1 - \beta_t)\hat{\xi}_{t-1}^i + \beta_t \xi_t^i$, where $\hat{\xi}_t^i$ is the running inner product maintained by the $t$-th mini-batch. This update process, which considers both the previous running inner product and the currently computed inner product, is equivalent to a *Simple Moving Average (SMA)* [Arce, 2004]. A detailed proof is provided in Appendix C. The complete orthogonalization algorithm is outlined in Algorithm 1.

## 4.2 Orthogonal Common Neighbor with Polynomial Filters

While Gram-Schmidt orthogonalization effectively reduces redundancy, it has a relatively high time complexity (see Appendix I for a detailed analysis). To address this, we explore alternative orthonormal bases for orthogonalization. For example, Chebyshev polynomials form an orthonormal basis and can be used as polynomial filters to process CNs.

Inspired by Wang and Zhang [2022], we propose using an orthonormal basis as polynomial filters to filter common neighbors. This can be expressed as: $OCN^K \approx \sum_{k=0}^{K} \alpha_k CN^k$, where $\alpha_k$ represents the coefficient of the $k$-th term in the polynomial filter basis. Although this approach compromises strict orthogonality, it reduces redundancy between $CN^k$ through spectral domain operations. To further reduce time complexity, we take the limiting case $T = 0$ and construct a diagonal matrix, applying consistent filtering operations to each edge signal of $CN^k$ within the same dimension.

By replacing inner product operations with weighted operations, we avoid the extensive computations and iterations required by Gram-Schmidt orthogonalization. This approach effectively adjusts the signal in the frequency domain and removes redundant information.

Building on this, we introduce Orthogonal Common Neighbor with Polynomial Filters (OCNP). We can select any popular orthogonal polynomial bases (e.g., Jacobi, Monomial, Chebyshev, or Bernstein). This operation can be viewed as passing the signal of each edge through a filter defined by the chosen polynomial, thereby adjusting the frequency characteristics of the signal. For a detailed analysis, please refer to Appendix J.

## 5 Normalization

A common issue is that, as the order of CNs increases, the high-order CNs of different nodes become more similar. To quantify this effect, we analyze the coefficient of variation of CNs at different orders across various nodes. As shown in Figure 2, the coefficient of variation decreases at higher orders, indicating that the high-order CNs of different nodes begin to overlap more frequently and that the similarity of higher-order neighborhood structures continually increases. This may lead to a loss of distinctiveness among nodes. In other words, as we consider more hops in the network, the feature representations of links become increasingly homogeneous, potentially undermining the performance of link prediction models.

This observation is consistent with the intuitive motivation behind our normalization trick. Specifically, when a node participates in a large number of walks, it will more frequently appear as a CN for other node pairs, causing the features of different links to become similar and thus leading to over-smoothing. Therefore, we propose a normalization trick similar to batch normalization [Ioffe and Szegedy, 2015] to mitigate this issue. We divide the coefficient of each node by the number of walks in which it

Figure 2: To demonstrate that the higher-order common neighbors of different node pairs become similar, we use the Cora dataset as an example. We calculate the coefficient of variation (CV), which is the ratio of the standard deviation to the mean, of the CNs of different nodes at the same order. High CV indicates low over-smoothing degree. Results shows that as the order increase, over-smoothing becomes more and more significant, but our method (yellow) can alleviate this problem.

participates to obtain *normalizedCN*. This normalization technique helps to reduce the influence of frequently appearing common neighbors, ensuring that nodes with a large number of high-order walk participations are not overly emphasized.

*normalizedCN* provides the following insight: *If $k$-hop CNs are less frequently shared among other node pairs, then these $k$-hop CNs carry greater significance in the relationship between the two nodes.* For example, if the more distant social circles in which two individuals indirectly participate include many people (i.e., are more mainstream), then the commonalities between these two individuals will be fewer. Conversely, if the social circle is more niche, it suggests a higher potential for a direct connection between the two individuals.

This compensates for the traditional Resource Allocation (RA) [Zhou et al., 2009] method, which ignores the potential contribution of $k$-hop CNs and fails to account for the higher-order structure of the graph. Next, we analyze this trick theoretically using a random graph model and prove its effectiveness. It can be seen that

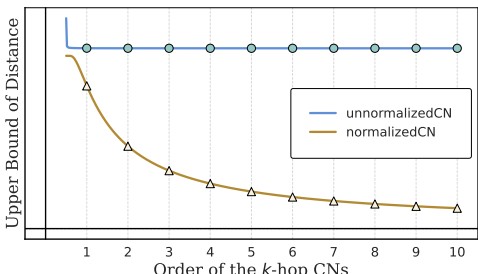

Figure 3: The impact of higher-order $k$-hop CNs on the upper bound of $d_{ij}$ is illustrated. $k$-hop CNs have no effect (blue line). The yellow line shows how the upper bound tightens with increasing $k$, which is the result obtained after introducing normalizedCN. With normalizedCN, the contribution of each $k$-hop CN is now $\sum_{c \in CN^k(i,j)} \frac{1}{|P_k(c)|}$ rather than just 1.

when the path or walk has a length of 1, the path count reduces to the node degree, and *normalizedCN* degenerates to RA.

## 5.1 Theoretical Justification for Normalization

In this section, we analyze common neighbor with and without normalization on random graphs and Barabási-Albert model [Albert and Barabási, 2002] to show that normalization leads to a better estimator of link existence. The latent space model [Hoff et al., 2002] is commonly applied to describe proximity in latent spaces, where nodes with similar locations in the latent space tend to share specific characteristics. We first discuss the $D$-dimensional Euclidean space (i.e., a space with curvature 0). We introduce a latent space model [Sarkar et al., 2011] for link prediction that describes a graph with $N$ nodes. All definitions and proofs in this section are in Appendix D.

To illustrate the effectiveness of normalization, we compare two link prediction heuristics: CN and normalized CN.

> **Definition 5.1.** *We define the **normalizedCN** between $(i, j)$ as a measure of their similarity with* Structural Feature *based on the contributions from all $k$-hop CNs. It is inversely proportional to the number of distinct node pairs for which it is also the $k$-hop common neighbor:*
> $$normalizedCN^k(i, j) = \sum_{c \in CN^k(i,j)} \frac{1}{|P_k(c)|},$$

where $CN^k(i, j)$ is the set of $k$-hop CNs of $(i, j)$, $|P_k(c)|$ is the total number of paths for the set of distinct node pairs in which node $c$ serves as a $k$-hop CN for those node pairs. Note that, even for the same node pair, there may be more than one path in which $c$ acts as the $k$-hop CN of $(i, j)$.

> **Theorem 5.2.** *When $k = 1$, normalizedCN$(i, j)$ degenerates into $RA(i, j)$. Specifically, for each $c \in CN^1(i, j)$, the following relationship holds: normalizedCN$^1(c, i, j) \cdot \frac{\binom{d(c)}{2}}{d(c)} = RA(c, i, j)$. The proof is trival.*

When there is no normalization, in the latent space we have the following Proposition 5.3. Here, $d_{ij}$ denotes the distance between node $i$ and node $j$ in the latent space. The smaller this distance is, the higher the probability that there exists a link between $i$ and $j$.

> **Proposition 5.3.** *(**Latent space distance bound with $k$-hop CNs**). For any $\delta > 0$, with probability at least $1 - \delta$, we have $d_{ij} \leq \sum_{n=0}^{M-2} r_n + 2\left((r_M^{\max})^2 - (\iota - \alpha)^{\frac{2}{D(2k-1)}}\right)^{\frac{1}{2}}$,*

where $\alpha = \frac{\sqrt{N \ln(1/2\delta)/2}}{N + \sqrt{-3N \ln \delta}}$, $\iota = \frac{\eta_{2k}(i,j)}{(N - \sqrt{-2N \ln \delta})^{2k-1}}$. And $r_M^{\max} = \max\{r_M\}(M \in \{1, \cdots, 2k-1\})$ is the maximum of the feature radius for the set of intermediate nodes in $D$ dimensional Euclidean space. $N$ is the number of nodes. $k$ represents the order of the $k$-hop CNs. $\eta_{2k}(i, j)$ is the number of $k$-hop CNs about $(i, j)$.

After applying our normalization trick, we have the following Proposition 5.4:

> **Proposition 5.4.** *(**Latent space distance bound with $k$-hop CNs after normalization**). We simply need to modify the overall contribution of $k$-hop CNs from $\eta_{2k}(i, j)$ to $\frac{\eta_{2k}(i,j)}{\sum_{c \in CN^k(i,j)} 1/|P_k(c)|}$. For any $\delta > 0$, with probability at least $1 - \delta$, we have*
> $$d_{ij} \leq \sum_{n=0}^{M-2} r_n + 2\sqrt{(r_M^{\max})^2 - \left(\left(\gamma\binom{\zeta}{2}\right)^{\frac{1}{D(k-1)}} \cdot \rho^N\right)^{\frac{2k-2}{2k-1}}}, \qquad (2)$$

where $\zeta$ is the maximum degree of all $k$-hop CNs of $(i, j)$ and $\rho \in [0, 1]$ and $\gamma = \left(\frac{\eta_{2k}(i,j)}{(N - \sqrt{-2N \ln \delta})^{2k-1}} - \alpha\right)$.

Next, we analyze the upper bound of $d(i, j)$ for these two cases. When there is no normalization, it is evident that as $N$ becomes large, $(1 - \alpha)$ approaches 1, so the order $k$ of the $k$-hop CNs has no effect. Additionally, the exponent term $\frac{2}{D(2k-1)}$ contains a large denominator $D$, meaning that even if $(1 - \alpha)$ does not approach 1, it would not significantly affect the result (as shown by the blue line in Figure 3). However, when we apply the normalization trick, the upper bound of $d_{ij}$ becomes tighter as $k$ increases (as shown by the yellow line in Figure 3). In fact, this aligns with the general belief that the effectiveness of incorporating higher-order $k$-hop CNs is indisputable [Wang et al.,

2024, Chamberlain et al., 2023, Mao et al., 2024, Yun et al., 2021]. Based on the above analysis, normalizedCN provides a tighter upper bound of $d(i,j)$ compared to CN and is more effective.

The above theoretical analysis was conducted on a random graph model. We extend the theoretical argument to the more realistic Barabási-Albert model [Albert and Barabási, 2002], and obtain the following Proposition 5.5 and Proposition 5.6 before and after introducing our path-based normalization, respectively. The detailed derivation process is provided in Appendix Q.

---

**Proposition 5.5.** *(Distance bound with $k$-hop CNs on Barabási-Albert model). For any $\delta > 0$, with probability at least $1 - \delta$, we have*

$$d_{ij} \leq 2k\left[\frac{1}{\alpha}\ln\left(\frac{2(N-2)}{\frac{(2N+1)!!}{2^N N!} + \frac{\sqrt{N \ln \delta^{-1}}}{4}} - 1\right) + \left[\frac{1}{NV(1)}\left(m\frac{(2N+1)!!}{2^N N!} + \sqrt{\frac{Nm^2}{2}\ln\delta^{-1}}\right)\right]^{\frac{1}{D}}\right],$$
(3)

*where $N = \#nodes$, $k$ represents the order of the $k$-hop CNs.*

---

**Proposition 5.6.** *(Distance bound with $k$-hop CNs on Barabási-Albert model after normalization). After introducing normalizedCN, for any $\delta > 0$, with probability at least $1 - \delta$, we have*

$$d_{ij} \leq 2k\left[\frac{1}{\alpha}\ln\left(\left[-\frac{n-2}{N-n-1}W\left(-\frac{N-n-1}{n-2}C^{\frac{1}{n-2}}\right)\right]^{-\frac{1}{k}} - 1\right) + E,$$
(4)

*where $W(\cdot)$ is **Lambert $W$ function**, $\zeta$ is the maximum degree of all $k$-hop CNs of $(i,j)$, the total number of paths of length $l$ between i and j is denoted as $\eta_l(i,j)$.*

$$C = \frac{1}{\binom{\zeta}{2}}\frac{D^{2k-1}}{\eta_{2k} - D^{2k-2}\frac{\sqrt{N \ln \delta^{-1}}}{4}}, E = \left[\frac{1}{NV(1)}\left(m\frac{(2N+1)!!}{2^N N!} + \sqrt{\frac{Nm^2}{2}\ln\delta^{-1}}\right)\right]^{\frac{1}{D}}\right]$$
(5)

*and $D$ is the maximum degree on the graph.*

---

Results show that without introducing normalizedCN, the upper bound of $s_{ij}$ is a monotonically increasing affine function with respect to $k$. After introducing normalizedCN, this upper bound gradually decreases as $k$ increases. This extends our theoretical analysis to more realistic scenarios.

# 6  Orthogonal Common Neighbor

Following the structure $0 \rightarrow OCN^0 \rightarrow \cdots \rightarrow OCN^k \rightarrow OCN^{k+1} \rightarrow \cdots$, we can naturally construct: $OCN^k \Rightarrow \sum_{u \in N^k(i) \cap N^k(j)} \text{MPNN}(u, A, X) = OCN^k \cdot h = OCN^k \cdot \text{MPNN}(A, x)$.

In particular, $OCN^0$ reflects the two nodes of the edge itself, so we have:

$$MPNN(N^0(i), A, x) \odot \text{MPNN}(N^0(j), A, x) = h_i \odot h_j = \text{MPNN}(i, A, x) \odot \text{MPNN}(j, A, x)$$
(6)

We naturally get our Orthogonal Common Neighbor (OCN) model:

$$\text{OCN}(i, j, A, X) = \text{MPNN}(i, A, X) \odot \text{MPNN}(j, A, X) + \sum_{k=1}^{K} \alpha_k \{\text{OCN}^k \cdot \text{MPNN}(A, X)\}_{ij}.$$
(7)

The complete algorithm is shown in Algorithm 2. The model architecture is detailed in Appendix G. According to Section 4.2, we obtain our Orthogonal Common Neighbor with Polynomial Filters (OCNP) by replacing the computation method of $OCN^k$ in Equation (7) from Algorithm 1 with Equation (73).

---

**Theorem 6.1.** *OCN is strictly more expressive than Graph Autoencoder(GAE), CN, RA, AA. Moreover, Neo-GNN BUDDY and NCN are not more expressive than OCN. The proof can be found in Appendix E.*

---

Table 1: Results on link prediction benchmarks. The format is average score ± standard deviation. OOM means out of GPU memory.

| | Cora | Citeseer | Pubmed | Collab | PPA | Citation2 | DDI |
|---|---|---|---|---|---|---|---|
| **Metric** | HR@100 | HR@100 | HR@100 | HR@50 | HR@100 | MRR | HR@20 |
| **CN** | 33.92±0.46 | 29.79±0.90 | 23.13±0.15 | 56.44±0.00 | 27.65±0.00 | 51.47±0.00 | 17.73±0.00 |
| **AA** | 39.85±1.34 | 35.19±1.33 | 27.38±0.11 | 64.35±0.00 | 32.45±0.00 | 51.89±0.00 | 18.61±0.00 |
| **RA** | 41.07±0.48 | 33.56±0.17 | 27.03±0.35 | 64.00±0.00 | 49.33±0.00 | 51.98±0.00 | 27.60±0.00 |
| **GCN** | 66.79±1.65 | 67.08±2.94 | 53.02±1.39 | 44.75±1.07 | 18.67±1.32 | 84.74±0.21 | 37.07±5.07 |
| **SAGE** | 55.02±4.03 | 57.01±3.74 | 39.66±0.72 | 48.10±0.81 | 16.55±2.40 | 82.60±0.36 | 53.90±4.74 |
| **SEAL** | 81.71±1.30 | 83.89±2.15 | 75.54±1.32 | 64.74±0.43 | 48.80±3.16 | 87.67±0.32 | 30.56±3.86 |
| **NBFnet** | 71.65±2.27 | 74.07±1.75 | 58.73±1.99 | OOM | OOM | OOM | 4.00±0.58 |
| **Neo-GNN** | 80.42±1.31 | 84.67±2.16 | 73.93±1.19 | 57.52±0.37 | 49.13±0.60 | 87.26±0.84 | 63.57±3.52 |
| **BUDDY** | 88.00±0.44 | 92.93±0.27 | 74.10±0.78 | 65.94±0.58 | 49.85±0.20 | 87.56±0.11 | 78.51±1.36 |
| **NCN** | 89.05±0.96 | 91.56±1.43 | 79.05±1.16 | 64.76±0.87 | 61.19±0.85 | 88.09±0.06 | 82.32±6.10 |
| **NCNC** | 89.65±1.36 | 93.47±0.95 | 81.29±0.95 | 66.61±0.71 | 61.42±0.73 | **89.12±0.40** | 84.11±3.67 |
| **PLNLP** | - | - | - | 70.59±0.29 | 32.38±2.58 | 84.92±0.29 | 90.88±3.13 |
| **OCN** | 89.82±0.91 | **93.62±1.30** | **83.96±0.51** | 72.43±3.75 | 69.79±0.85 | 88.57±0.06 | 97.42±0.34 |
| **OCNP** | **90.06±1.01** | 93.41±1.02 | 82.32±1.21 | 67.74±0.16 | **74.87±0.94** | 87.06±0.27 | **97.65±0.38** |

In our ablation study, when incorporating 3-hop common neighbors (3-hop CN), the model's performance did not show significant improvement and instead exhibited instability and increased fluctuations during training. This suggests that higher-order neighbors may introduce redundant information, affecting the model's stability and generalization ability. Therefore, despite the possibility of selecting more orthogonal bases, and considering training resources and model stability, we prefer to use $OCN^1$ and $OCN^2$ to ensure efficiency on large-scale graph datasets. The analysis of $\alpha_1$ and $\alpha_2$ along with a discussion highlighting differences from NCN can be found in Appendix N.

# 7 Experiment

In this section, we present a comprehensive evaluation of the performance of OCN. The full experimental setup is provided in Appendix F and Appendix H.

For our evaluation, we utilize seven well-known real-world datasets for link prediction. Three of these datasets come from Planetoid's citation networks: Cora, Citeseer, and Pubmed [Yang et al., 2016]. The remaining datasets are sourced from the Open Graph Benchmark (OGB) [Hu et al., 2020], including ogbl-collab, ogbl-ppa, ogbl-citation2, and ogbl-ddi. Detailed statistics and dataset splits are provided in Appendix F.

## 7.1 Evaluation on Real-World Datasets

In our evaluation on real-world datasets, we adopted a series of baseline methods, including traditional heuristic approaches such as CN [Barabási and Albert, 1999], RA [Zhou et al., 2009], and AA [Adar and Adamic, 2003], as well as GAE models such as GCN [Kipf and Welling, 2017] and SAGE [Hamilton et al., 2017], SF-then-MPNN models, including SEAL [Zhang and Chen, 2018] and NBFNet [Zhu et al., 2021], as well as SF-and-MPNN models such as Neo-GNN [Yun et al., 2021] and BUDDY [Chamberlain et al., 2023]. We also compared with models that adopt the same MPNN-then-SF architecture, including NCN and NCNC [Wang et al., 2024]. In ad-

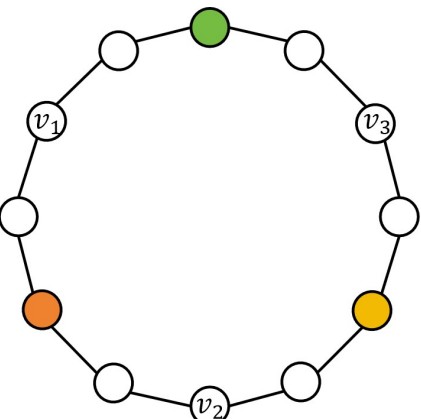

Figure 4: White, green, orange, and yellow represent node features 0, 1, 2, and 3, respectively. $v_2$ and $v_3$ are symmetric, and GAE cannot distinguish $(v_1, v_2)$ and $(v_1, v_3)$. With node features ignored, $(v_1, v_2)$ and $(v_1, v_3)$ are symmetric, so CN, RA, AA, Neo-GNN, and BUDDY cannot distinguish them. NCN also degenerates into GAE, so it also cannot. However, $(v_1, v_2)$ and $(v_1, v_3)$ have different 2-hop CNs, which allows OCN to distinguish them.

Table 2: Ablation study on link prediction benchmarks. The format is average score ± standard deviation. OOM means out of GPU memory.

| | Cora | Citeseer | Pubmed | Collab | PPA | Citation2 | DDI |
|---|---|---|---|---|---|---|---|
| Metric | HR@100 | HR@100 | HR@100 | HR@50 | HR@100 | MRR | HR@20 |
| OCN-Orth | 88.62±1.16 | 92.04±1.53 | 80.95±1.14 | 64.64±0.45 | 18.39±5.94 | 74.53±4.31 | 36.08±9.36 |
| OCN-normalizedCN | 89.11±1.45 | 92.25±1.46 | 81.35±0.98 | 67.30±0.33 | 59.94±2.65 | 86.33±0.17 | 97.56±0.43 |
| OCN-CAT | 88.63±0.99 | 92.52±1.37 | 81.12±0.39 | 63.20±0.80 | 34.30±6.55 | 87.34±0.23 | 24.30±3.11 |
| OCN-Linear | 87.73±1.23 | 90.60±1.24 | 79.71±1.08 | 64.58±0.56 | 6.91±3.34 | 88.56±0.13 | 84.89±2.74 |
| OCN-sfANDmpnn | 89.07±1.35 | 91.98±1.32 | 80.27±1.01 | 68.75±0.34 | 59.30±0.54 | OOM | 80.13±12.65 |
| OCN-3 | 85.84±2.23 | 88.60±4.59 | 69.56±4.66 | 72.28±1.33 | OOM | 82.88±3.02 | 89.19±3.21 |
| OCN-SPD | 89.17±1.15 | 91.97±1.60 | 80.69±1.33 | 62.58±0.91 | 44.37±2.19 | 86.99±0.68 | 96.82±0.16 |
| OCN | 89.82±0.91 | 93.62±1.30 | 83.96±0.51 | 72.43±3.75 | 69.79±0.85 | 88.57±0.06 | 97.42±0.34 |
| OCNP-Filter | 88.73±1.36 | 92.18±2.65 | 81.40±0.88 | 63.09±1.75 | 30.86±1.03 | 86.96±0.19 | 27.27±4.17 |
| OCNP-CAT | 88.05±1.53 | 91.71±1.66 | 81.61±0.73 | 63.89±0.39 | 28.11±2.01 | 86.98±0.44 | 26.13±4.85 |
| OCNP-Linear | 87.68±1.41 | 90.93±1.79 | 80.29±0.90 | 60.89±0.91 | 12.82±1.37 | 87.10±0.28 | 49.48±0.34 |
| OCNP-sfANDmpnn | 88.95±0.96 | 92.36±1.40 | 79.35±0.63 | 66.90±1.29 | 57.45±0.89 | OOM | 96.86±0.11 |
| OCNP-3 | 88.17±1.44 | 91.60±2.04 | 79.03±0.99 | 59.31±0.57 | OOM | OOM | 91.31±2.79 |
| OCNP-SPD | 89.08±1.37 | 91.36±1.85 | 80.25±0.47 | 66.24±0.51 | 52.93±1.14 | 87.27±0.74 | 97.33±0.95 |
| OCNP | 90.06±1.01 | 93.41±1.02 | 82.32±1.21 | 67.74±0.16 | 74.87±0.94 | 87.06±0.27 | 97.65±0.38 |

dition, we also selected the strong baseline PLNLP [Wang et al., 2021], which uses training tricks. Furthermore, we compared it with GIDN [Wang et al., 2022], which also utilizes training tricks, as well as several node embedding methods. The detailed comparison results can be found in Appendix K. The baseline results are derived from [Wang et al., 2024]. Our model is OCN, and its architecture is detailed in Appendix G.

The experimental results, as shown in Table 1, demonstrate that OCN outperforms all baselines on all datasets except Citeseer and Citation2. Compared to the best-performing NCNC, OCN is only 0.6% behind on Citeseer and Citation2, but OCN still outperforms all baselines except NCNC on these two datasets. On the remaining five datasets, OCN improves by an average of 7.7% over NCNC. Across the seven datasets, OCN improves by an average of 7.2% over NCN and by an average of 12.4% over BUDDY. These excellent results undoubtedly prove the superior expressiveness of our OCN model. Furthermore, on the ogbl-ppa dataset, OCN surpassed the large-scale GraphGPT [Zhao et al., 2024] (0.6876 ± 0.0067), and on the ogbl-ddi dataset, it achieved 97.42, significantly outperforming the strongest baseline, NCNC. OCNP not only maintains the overall performance of OCN but also significantly reduces computational complexity, while achieving substantially superior results compared to NCNC on the ogbl-ppa. The famous baseline models' average rankings are: BUDDY: 4.7, NCN: 4.29, NCNC: 2.71, while our models' average rankings are: OCN: **1.57**, OCNP: **2.28**.

Additionally, we conducted a comparison between OCN, OCNP, and other Link Prediction Models, as detailed in Section K. Our models consistently demonstrated superior performance when compared to both node embedding-based methods and models that incorporated various training tricks, regardless of whether these tricks were applied or not. The results show that, even with the introduction of such training techniques, our models still maintain an edge in terms of prediction accuracy and efficiency. Furthermore, we have also provided an ablation study on the MPNN used in both OCN and OCNP in Section L. This study offers deeper insights into the specific factors in MPNN that enhance the performance.

## 7.2 Ablation Analysis

To assess the effectiveness of the OCN and OCNP design, we conducted a comprehensive ablation analysis, as shown in Table 2. By eliminating the influence of normalizedCN on the $k$-hop CN weights, we derive OCN-Orth. Similarly, removing the effect of *normalizedCN* from OCNP results in OCN-Filter. Furthermore, keeping the impact of *normalizedCN* while removing the orthogonalization process in OCN and the filtering mechanism in OCNP leads to OCN-normalizedCN. Additionally, we modify both models in Equation (7) as follows: $\mathrm{MPNN}(i, A, X) \odot \mathrm{MPNN}(j, A, X) \big\| \sum_{k=1}^{K} \alpha_k \, \mathrm{OCN}^k \cdot \mathrm{MPNN}(A, X)$, which results in OCN-CAT and OCNP-CAT. The experimental results for OCN-CAT and OCNP-CAT show notably lower performance, especially on PPA. A detailed analysis of the significant performance gap resulting from changing the aggregation strategy from summation to concatenation can be found in Appendix O. To investigate the impact

of the nonlinear layers in our models, we remove the nonlinear layers from both OCN and OCNP, yielding OCN-Linear and OCNP-Linear, respectively. To verify that employing the *MPNN-then-SF* paradigm better addresses the deficiencies mentioned earlier, we construct two variants utilizing the *SF-and-MPNN* framework. Finally, when incorporating 3-hop CNs, the models are referred to as OCN-3 and OCNP-3. When we use SPD-based higher-order common neighbors, we obtain OCN-SPD and OCNP-SPD.

Beyond thoroughly verifying the significant role of each component in our models, an intriguing experimental observation is that removing the nonlinear layers does not lead to a drastic performance drop, except for ogbl-ppa, where performance collapses entirely. While the overall performance gap on smaller-scale graph datasets remains relatively minor, it is important to highlight that OCN and OCNP are originally designed to achieve substantial improvements on large-scale graphs, whereas their benefits on smaller graphs are naturally less pronounced. This is likely because extensively leveraging high-order CNs may introduce more uninformative information in smaller-scale graphs. Furthermore, incorporating 3-hop CNs not only imposes a substantial increase in computational cost and time but also contributes to greater instability in model performance. This instability manifests as significant oscillations in the loss curve, which could potentially stem from the increased norm of the model. MPNN may still implicitly learn information from higher-order neighbors. Using SPD-based higher-order common neighbors also results in a performance decline, as we have analyzed, due to information loss.

### 7.3   Scalability

We compare the inference time and GPU memory usage on the ogbl-collab dataset in Figure 5. Both OCN and NCN exhibit similar computational overhead, as they only need to run the MPNN model once. However, in contrast, SEAL shows a significantly higher computational overhead, particularly as the batch size increases. This is because SEAL needs to rerun the MPNN model for each individual target link, which leads to a substantial increase in inference time with larger batch sizes. When it comes to GPU memory consumption, OCN generally requires more memory than NCN. OCNP tends to outperform OCN in both inference speed and memory consumption.

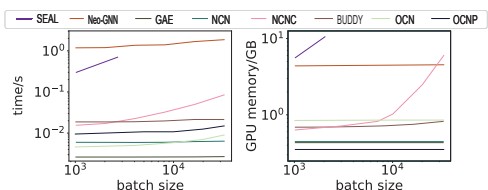

Figure 5: Inference time and GPU memory on ogbl-collab. The measured process includes pre-processing and predicting one batch of test links. As shown in Appendix I, the relation between time $y$ and batch size $t$ is $y = B + Ct$.

We also conducted similar tests on a variety of other datasets. The results are shown in Figure 9. We observe patterns that are quite similar to the ones seen in the ogbl-collab dataset, as presented in Figure 5. Specifically, both OCN and OCNP generally show better scalability than Neo-GNN, handling larger datasets more efficiently. On the other hand, SEAL continues to demonstrate the poorest scalability among the models tested. In terms of memory usage, the overhead of OCN is either comparable to or slightly higher than that of NCN, depending on the dataset.

## 8   Conclusion

We propose a novel method called **OCN** for the link prediction task and effectively alleviates two key issues. **OCN** demonstrates significant performance improvements across multiple datasets. Furthermore, to further reduce the time complexity of **OCN**, we introduce **OCNP**. The work presented in this study offers a new perspective on the link prediction task and provides valuable insights for future research on handling higher-order neighborhood information in large-scale graphs.

## Acknowledgments and Disclosure of Funding

This work is supported by the National Key R&D Program of China (2022ZD0160300) and the National Natural Science Foundation of China (62276003).

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

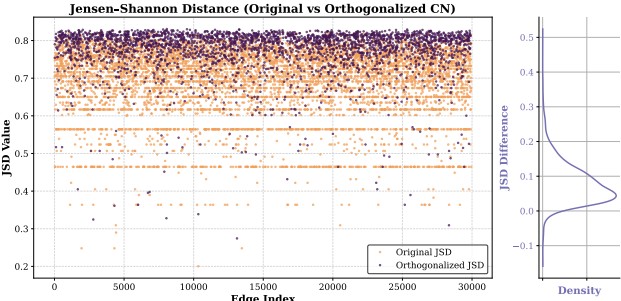

Figure 6: This figure demonstrates that orthogonalization reduces redundancy in higher-order common neighbors, as shown by the increased and concentrated JSD values, making the representations more independent and distinguishable.

# A    Evaluating the Effectiveness of Orthogonalization

The orthogonalization process removes redundant information at the matrix granularity. However, for the point pairs $(i, j)$ that we are interested in, we need to measure the similarity of the distribution of information between different $k$-hop CNs at different orders for this point pair. By comparing the difference in similarity at the edge granularity before and after the orthogonalization process, we can objectively assess the effectiveness of orthogonalization in eliminating redundant information at a finer granularity. We introduce the Jensen-Shannon Divergence (JSD) as a metric, with the core goal being to verify whether orthogonalization can effectively reduce the similarity of the common neighbor matrices between different orders for each edge.

Let $p_e$ and $q_e$ represent the vector corresponding to edge $e = (i, j)$ in the matrices of $CN^\alpha$ and $CN^\beta$, respectively. We will normalize them: $\tilde{p}_e = \frac{p_e}{\sum_k p_{e,k}}, \quad \tilde{q}_e = \frac{q_e}{\sum_k q_{e,k}}$, and then substitute them into the definition of Jensen-Shannon distance:

$$D_{JS}(p_e, q_e) = \frac{1}{2}\left(D_{KL}(\tilde{p}_e \parallel \tilde{m}_e) + D_{KL}(\tilde{q}_e \parallel \tilde{m}_e)\right), \tag{8}$$

where $\tilde{m}_e$ and $\tilde{q}_e$ represent the average of $\tilde{p}_e$ and $\tilde{q}_e$, which are defined as $\tilde{m}_e = \frac{1}{2}(\tilde{p}_e + \tilde{q}_e)$.

As shown in the left part of Figure 6, taking the Collab dataset's $CN^1$ and $CN^2$ as an example, the JSD values between each edge's distribution of $CN^1$ and $CN^2$ are represented by the orange points. The JSD values after orthogonalization (purple points) are generally higher and more concentrated, indicating that the orthogonalization process has successfully transformed the higher-order CN matrices into more independent and distinguishable representations for each node pair $(i, j)$. The density curve on the right reflects the difference before and after orthogonalization for each edge. It is evident that the JSD values for the majority of $(i, j)$ pairs have increased after orthogonalization.

# B    Detailed Representation and Meaning of High-order Common Neighbors

According to Equation (1), assuming there exist $k$-hop CNs for the two endpoints of interest, the following cases arise:

For $k$-hop CNs, if $l = 2\lambda$ (even), the path contains exactly one $k$-hop CN (each endpoint is $k$-hops away from the CN). However, there might be more than one such $k$-hop path. In this case, we construct $\omega$ as shown below:

$$\begin{cases} \omega_{i_0 \cdots \hat{i}_k \cdots i_{2k}} = A^k_{i_0 i_k} A^k_{i_{2k} i_k} & \forall i_k \in V \\ \omega_{i_0} \cdots \hat{i}_p \cdots i_{2k} = 0 & p \in [0, 2k], p \in \mathbb{Z}, p \neq k \end{cases} \tag{9}$$

where $A_{ab}^k$ represents the element in the $a$-th row and $b$-th column of the $k$-th power of the adjacency matrix. If $l = 2\lambda - 1$ (odd), the path contains two $k$-hop CNs. we construct $\omega$ as shown below:

$$
\begin{cases}
\omega_{i_0\cdots\hat{i}_{k-1}\cdots i_{2k-1}} = A_{i_0 i_{k-1}}^{k-1} A_{i_{2k-1} i_{k-1}}^k, & \forall i_{k-1} \in V \\
\omega_{i_0\cdots\hat{i}_k\cdots i_{2k-1}} = A_{i_0 i_k}^k A_{i_{2k-1} i_k}^{k-1}, & \forall i_k \in V \\
\omega_{i_0\cdots\hat{i}_p\ldots i_{2k-1}} = 0, & \\
p \in [0, 2k-1],\, p \in \mathbb{Z},\, p \neq k-1,\, p \neq k
\end{cases}
\tag{10}
$$

As defined above which can be summarized plainly as: For each order $k$, we maintain three $(h, n)$-shaped matrices. For example: For 4-order CN: Possible path length to $(u, v)$ combinations are $(4, 4)$, $(3, 4)$, or $(4, 3)$. For 7-order CN: Possible combinations are $(7, 7)$, $(6, 7)$, or $(7, 6)$.

Take a matrix representing $(3, 4)$ distances as example: Element $z$ at position $(x, y)$ indicates: For the $x$-th edge $(u, v)$ in the batch, node $y$ has exactly $z$ paths to $(u, v)$ with lengths precisely $(3, 4)$. If $z = 0$: Node $y$ cannot reach $(u, v)$ via any $(3, 4)$-length paths $\rightarrow$ it's not a 4-order CN under $(3, 4)$ definition. It might still qualify as a 4-order CN under $(4, 4)$ or $(4, 3)$ definitions. If none apply: Node y is not a 4-order CN for $(u, v)$ at all.

## C   Proof of Convergence to Global Orthogonality.

*Proof.*

$$
\hat{\xi}_t = (1 - \beta_t)\,\hat{\xi}_{t-1} + \beta_t \xi_t \tag{11}
$$

$$
= \prod_{i=1}^{t} (1 - \beta_i)\,\hat{\xi}_0 + \sum_{i=1}^{t} \prod_{j=i}^{t} (1 - \beta_{j+1})\beta_j \xi_i \tag{12}
$$

$$
= \frac{1}{t+1}\hat{\xi}_0 + \sum_{i=1}^{t} \frac{1}{t+1}\xi_i \tag{13}
$$

$$
= \frac{1}{t+1} \sum_{i=0}^{t} \xi_i \Leftrightarrow SMA \tag{14}
$$

The inner product $\xi_t^i$ of each mini-batch is an independent and identically distributed (i. i. d. ) random variable with a finite expected value $\mathbb{E}[\xi_t^i] = \xi^i = \langle CN^k, OCN^i \rangle$ which represents the global inner product of the $k$-th $CN$ vector with the $i$-th $OCN$ vector over the entire graph. As $t \rightarrow \infty$, the running average of the inner product converges to its expectation.

$\square$

## D   Related Derivation of normalizedCN.

**Definition D.1.** *we introduce a latent space model [Sarkar et al., 2011] for link prediction that describes a graph with $N$ nodes, each associated with a location in the space:*

$$
P(i \sim j | d_{ij}) = \begin{cases} \frac{1}{1 + e^{\alpha(d_{ij} - \max\{r_i, r_j\})}} & \text{if } d_{ij} \leq \max\{r_i, r_j\} \\ 0 & \text{if } d_{ij} > \max\{r_i, r_j\} \end{cases}
\tag{15}
$$

where $P(i \sim j | d_{ij})$ denotes the probability of forming an edge between nodes $i$ and $j$ . $d_{ij}$ represents the latent distance between nodes, indicating the likelihood of a link forming between them. The model has two parameters, $\alpha$ and $r$, where $\alpha > 0$ controls the steepness of the function. To ease the analysis, we set $\alpha = +\infty$. $r_i$ is a connecting threshold parameter corresponding to node $i$. With $\alpha = +\infty$, we have

$$
\frac{1}{1 + e^{\alpha(d_{ij} - \max\{r_i, r_j\})}} = 0 \quad \text{if } d_{ij} > \max\{r_i, r_j\} \tag{16}
$$

otherwise (16) equals to 1.

The nodes are distributed uniformly across a $D$-dimensional Euclidean space, with each node having an associated radius $r$ and a corresponding volume $V(r)$. The probability of establishing a connection between any two nodes $i$ and $j$, denoted as $P(i \sim j)$, is influenced by both the radii $(r_i, r_j)$ and the distance $d_{ij}$ between them.

1. The volume of a ball with radius $r$ is expressed as $V(r) = V(1)r^D$, where $V(r)$ refers to the volume of a ball with radius $r$ and $V(1)$ is the volume of a unit-radius hypersphere.

2. The degree of a node $i$, represented by $\mathrm{Deg}(i)$, is proportional to the volume $V(r_i)$ of the ball corresponding to the node's radius, and is given by $\mathrm{Deg}(i) = NV(r_i)$, where $N$ represents the total number of nodes.

The likelihood of a link between two nodes $i$ and $j$ forming as a function of their distance $d_{ij}$ is given by the following logistic expression:

$$P(i \sim j \mid d_{ij}) = \frac{1}{1 + e^{\alpha(d_{ij} - \max\{r_i, r_j\})}}, \tag{17}$$

where $P(i \sim j \mid d_{ij})$ signifies the probability of a connection between nodes $i$ and $j$, $\alpha > 0$ governs the steepness of the transition, and $\max\{r_i, r_j\}$ defines the critical radius at which the likelihood drops sharply.

In order to ensure the proper normalization of probabilities, we assume that all the nodes are contained within a hypersphere of unit volume in the $D$-dimensional space. The maximum allowable radius, denoted $r_{\mathrm{MAX}}$, is chosen such that:

$$V(r_{\mathrm{MAX}}) = V(1)r_{\mathrm{MAX}}^D = 1, \quad \text{which implies that } r_{\mathrm{MAX}} = \left(\frac{1}{V(1)}\right)^{1/D}. \tag{18}$$

For any pair of nodes $i$ and $j$, the volume of the intersection of the balls with radii $r_i$ and $r_j$, which are separated by a distance $d_{ij}$, is represented by $A(r_i, r_j, d_{ij})$. The intersection volume is bounded as follows using the properties of hyperspheres:

$$\left(\frac{r_i + r_j - d_{ij}}{2}\right)^D \leq \frac{A(r_i, r_j, d_{ij})}{V(1)} \leq \left((r_{ij}^{\max})^2 - \left(\frac{d_{ij}}{2}\right)^2\right)^{D/2}, \tag{19}$$

where $r_{ij}^{\max} = \max\{r_i, r_j\}$. This relation connects the intersection volume $A(r_i, r_j, d_{ij})$ to the distance $d_{ij}$ through the volume of the unit hypersphere.

**Definition D.2.** *(simple path and set) Given nodes $i$ and $j$ in a graph $G(V, E)$, a simple path of length $\ell$ from $i$ to $j$ is defined as a sequence $\mathrm{path}(i, k_1, k_2, \ldots, k_{\ell-2}, j)$, where $i \sim k_1 \sim k_2 \sim \cdots \sim k_{\ell-2} \sim j$, and $S_\ell(i, j)$ represents the set of all such possible paths, where each intermediate node $k_1, k_2, \ldots, k_{\ell-2}$ belongs to the set of vertices $V$.*

*Let $Y(i, k_1, k_2, \ldots, k_{\ell-2}, j)$ be a random variable which takes the value 1 if the path $(i, k_1, k_2, \ldots, k_{\ell-2}, j)$ belongs to $S_\ell(i, j)$, and 0 otherwise. The total number of paths of length $\ell$ between $i$ and $j$, denoted as $\eta_\ell(i, j)$, is then given by:*

$$\eta_\ell(i, j) = \sum_{k_1, \ldots, k_{\ell-2} \in S_\ell(i, j)} Y(i, k_1, \ldots, k_{\ell-2}, j \mid d_{ij}) \tag{20}$$

**Lemma D.3.**

$$\Delta < N\left(1 - \sqrt{-2\ln\delta/N}\right) \quad \text{with probability at least } 1 - \delta \tag{21}$$

$$\Delta > N\left(1 + \sqrt{-3\ln\delta/N}\right) \quad \text{with probability at least } 1 - \delta \tag{22}$$

*where $\Delta$ is the maximum degree.*

*Proof.* The degree $\mathrm{Deg}(k)$ of any node $k$ is a binomial random variable with expectation $\mathbb{E}[\mathrm{Deg}(k)] = NV(r_k)$, where $V_{r_k}$ is the volume of a hypersphere of radius $r_k$. Thus, using the Chernoff bound [Lugosi, 2008],

$$\mathrm{Deg}(k) < NV(r_k)\left(1 - \sqrt{-2\ln\delta/NV(r_k)}\right) \quad \text{holds with probability at least } 1 - \delta \qquad (23)$$

$$\mathrm{Deg}(k) > NV(r_k)\left(1 + \sqrt{-3\ln\delta/NV(r_k)}\right) \quad \text{holds with probability at least } 1 - \delta \qquad (24)$$

Applying the union bound on all nodes yields the desired proposition, i. e. ,

$$\Delta < NV(r_{\mathrm{MAX}})\left(1 - \sqrt{-2\ln\delta/NV(r_{\mathrm{MAX}})}\right) = N\left(1 - \sqrt{-2\ln\delta/N}\right). \qquad (25)$$

$$\Delta > NV(r_{\mathrm{MAX}})\left(1 + \sqrt{-3\ln\delta/NV(r_{\mathrm{MAX}})}\right) = N\left(1 + \sqrt{-3\ln\delta/N}\right). \qquad (26)$$

$\square$

**Lemma D.4.** *For any graph with maximum degree $\Delta$, we have:*

$$\eta_\ell(i,j) \leq \Delta^{\ell-1}. \qquad (27)$$

*Proof.* This can be demonstrated using a straightforward inductive approach. When the graph is represented by its adjacency matrix $M$, the number of paths of length $\ell$ between nodes $i$ and $j$ is given by $M^\ell(i,j)$. It is clear that $M^2(i,j)$ can be at most $\Delta$, which occurs when both $i$ and $j$ have degree $\Delta$, and their respective neighbors form a perfect matching. Assuming the inductive hypothesis holds for all $m < \ell$, we obtain the following:

$$M^\ell(i,j) = \sum_p M(i,p)M^{\ell-1}(p,j) \leq \Delta^{\ell-2}\sum_p M(i,p) \leq \Delta^{\ell-1}. \qquad (28)$$

$\square$

**Lemma D.5.** *For $\ell < \Delta$,*

$$\left|\eta_\ell\left(i,j \mid X_1,\ldots,X_p,\ldots,X_N\right) - \eta_\ell\left(i,j \mid X_1,\ldots,\tilde{X}_p,\ldots X_N\right)\right| \leq \Delta^{\ell-2} \qquad (29)$$

*Proof.* Consider all paths where $p$ is $m$ hops from $i$ (and hence $\ell-m$ hops from $j$). From Theorem D.4, the number of such paths can be at most

$$\Delta^{m-1} \cdot \Delta^{\ell-m-1} = \Delta^{\ell-2} \qquad (30)$$

$\square$

With the groundwork laid above, we can now proceed to prove our main result in Theorem 5.3. Here, we restate it once again:

**Proposition D.6.** *(Latent space distance bound with k-hop CNs). For any $\delta > 0$, with probability at least $1 - \delta$, we have*

$$d_{ij} \leq \sum_{n=0}^{M-2} r_n +$$

$$2\left((r_M^{\max})^2 - \left(\frac{\eta_{2k}(i,j)}{(N - \sqrt{-2N\ln\delta})^{2k-1}} - \alpha\right)^{\frac{2}{D(2k-1)}}\right)^{\frac{1}{2}} \qquad (31)$$

$$\leq \sum_{n=0}^{M-2} r_n + 2\sqrt{(r_M^{\max})^2 - (1-\alpha)^{\frac{2}{D(2k-1)}}}$$

*where $\alpha = \sqrt{N\ln(1/2\delta)/2}/\left(N + \sqrt{-3N\ln\delta}\right)$. $r_M^{\max} = \max\{r_M\}(M \in \{1,\cdots,2k-1\})$ is the maximum of the feature radius for the set of intermediate nodes in $D$ dimensional Euclidean space.*

*N is the number of nodes. $k$ represents the order of the $k$-hop CNs. $\eta_{2k}(i,j)$ is the number of $k$-hop CNs about $(i,j)$. (The above only shows the case where the $k$-hop CNs are in symmetric positions, as in (9). The asymmetric case, as in (10), is very similar. )*

*Proof.* Define $P_\ell(i,j)$ as the probability of observing an $\ell$-hop path between points $i$ and $j$. Next, we compute the expected number of $\ell$-hop paths.

Consider an $\ell$-hop path between $i, j$, for clarity of notation, let us denote the distances $d_{i,k_1}, d_{k_1,k_2}$, etc. by $a_1, a_2$, up to $a_{\ell-1}$ and radius $r_i, r_{k_1}, \ldots, r_j$ by $r_0, r_1, \ldots, r_{\ell-1}$. We also denote the distances $d_{jk_1}, d_{jk_2}, \ldots$ by $d_1, d_2, \ldots, d_{\ell-1}$. Note $r'_j = \max(r_{j-1}, r_j), j \in \{1, 2, \ldots, \ell-1\}$.

From the triangle inequality,

$$d_{\ell-3} \leq a_{\ell-2} + a_{\ell-1} \leq r_{\ell-2} + r_{\ell-1}, \tag{32}$$

and by induction,

$$d_k \leq \sum_{m=k+1}^{\ell} r_m. \tag{33}$$

Similarly,

$$d_1 \geq (d_{ij} - a_1)_+ \geq (d_{ij} - r_i)_+, \tag{34}$$

and by induction,

$$d_k \geq \left(d_{ij} - \sum_{n=0}^{k-1} r_n\right)_+. \tag{35}$$

**Case 1 (Symmetric case):** The K-order common neighbor is located at the midpoint of the path.

$$
\begin{aligned}
P_{2k}(i,j) &= P\left(i \sim k_1 \sim \ldots \sim k_{2k-1} \sim j \mid d_{ij}\right) \\
&= P\left(a_1 \leq r'_1 \cap \ldots \cap a_{2k} \leq r'_{2k-1} \mid d_{ij}\right) \\
&= \int_{d_1,\ldots,d_{2k-2}} P\left(a_1 \leq r'_1, \ldots, a_{2k-1} \leq r'_{2k-1}, d_1, \ldots, d_{2k-2} \mid d_{ij}\right) \\
&= \int_{d_{2k-2}=(d_{ij}-\sum_{n=0}^{2k-3} r_n)_+}^{r_{2k-1}+r_{2k}} \cdots \int_{d_1=(d_{ij}-r_0)_+}^{\sum_{m=2}^{2k} r_m} P\left(a_1 \leq r'_1, d_1 \mid d_{ij}\right) \ldots P\left(a_{2k-1} \leq r'_{2k-1}, a_{2k} \leq r'_{2k} \mid d_{2k-2}\right) \\
&\leq A\left(r'_1, \sum_{m=2}^{2k} r_m, d_{ij}\right) \times A\left(r'_2, \sum_{m=3}^{2k} r_m, (d_{ij}-r_0)_+\right) \times \ldots \times A\left(r'_{2k-1}, r_{2k}, \left(d_{ij} - \sum_{n=0}^{2k-3} r_n\right)_+\right) \\
&\leq \prod_{p=1}^{2k-1} A\left(r'_p, \sum_{m=p+1}^{2k} r_m, \left(d_{ij} - \sum_{n=0}^{p-2} r_n\right)_+\right)
\end{aligned}
\tag{36}
$$

$$E\left[\eta_{2k}(i,j)\right] \leq \Delta^{2k-1} \left[\prod_{p=1}^{2k-1} A\left(r'_p, \sum_{m=p+1}^{2k} r_m, \left(d_{ij} - \sum_{n=0}^{p-2} r_n\right)_+\right)\right] \tag{37}$$

**Case 2 (Asymmetric case):** The K-order common neighbor is located at positions on the path, at a distance of k-1 hops and k hops from the endpoints, or at positions at a distance of k hops and k-1 hops from the endpoints.

Similarly,

$$E\left[\eta_{2k-1}(i,j)\right] \leq 2\Delta^{2k-2} \left[\prod_{p=1}^{2k-2} A\left(r'_p, \sum_{m=p+1}^{2k-1} r_m, \left(d_{ij} - \sum_{n=0}^{p-2} r_n\right)_+\right)\right] \tag{38}$$

Due to the high similarity between the two cases above, in the following analysis, we will focus only on the symmetric case.

Through empirical Bernstein bounds [McDiarmid et al., 1989], we have: For any $t > 0$,

$$\Pr\left(f(X_1, \ldots, X_n) - \mathbb{E}[f(X_1, \ldots, X_n)] \geq t\right) \leq \exp\left(-\frac{2t^2}{\sum_{i=1}^n c_i^2}\right). \tag{39}$$

Back to our main proof, so we have:

$$\eta_{2k}(i,j) \leq E\left[\eta_{2k}(i,j)\right] + \Delta^{2k-2}\sqrt{\frac{N\ln(1/2\delta)}{2}}$$

$$\leq \Delta^{2k-1}\left[\prod_{p=1}^{2k-1} A\left(r_p', \sum_{m=p+1}^{2k} r_m, \left(d_{ij} - \sum_{n=0}^{p-2} r_n\right)_+\right) + \frac{\sqrt{\frac{N\ln(1/2\delta)}{2}}}{\Delta}\right]$$

$$\leq \left[\prod_{p=1}^{2k-1} A\left(r_p', \sum_{m=p+1}^{2k} r_m, \left(d_{ij} - \sum_{n=0}^{p-2} r_n\right)_+\right) + \frac{\sqrt{\frac{N\ln(1/2\delta)}{2}}}{N + \sqrt{-3N\ln\delta}}\right] \cdot (N - \sqrt{-2N\ln\delta})^{2k-1} \tag{40}$$

which can be rewritten as:

$$\eta_{2k}(i,j) \leq c(N,\delta,k)\prod_{p=1}^{2k-1} A\left(r_p', \sum_{m=p+1}^{2k} r_m, \left(d_{ij} - \sum_{n=0}^{p-2} r_n\right)_+\right) + b(N,\delta,k) \tag{41}$$

where $c(N,\delta,k) = (N - \sqrt{-2N\ln\delta})^{2k-1}$ and $b(N,\delta,k) = \frac{\sqrt{\frac{N\ln(1/2\delta)}{2}}}{N+\sqrt{-3N\ln\delta}} \cdot (N - \sqrt{-2N\ln\delta})^{2k-1}$.

Note $r_p^{\max} = \max\{r_p', \sum_{m=p+1}^{2k} r_m\}$, we have:

$$\eta_{2k}(i,j) \leq c(N,\delta,k)\prod_{p=1}^{2k-1}\left((r_p^{\max})^2 - \left(\frac{\left(d_{ij} - \sum_{n=0}^{p-2} r_n\right)_+}{2}\right)^2\right)^{D/2} + b(N,\delta,k)$$

$$= c(N,\delta,k)\left(\prod_{p=1}^{2k-1}\left[(r_p^{\max})^2 - \left(\frac{d_{ij} - \sum_{n=0}^{p-2} r_n}{2}\right)^2\right]\right)^{D/2} + b(N,\delta,k)$$

$$\leq c(N,\delta,k)\left(\prod_{p=1}^{2k-1}\left[(r_M^{\max})^2 - \left(\frac{d_{ij} - \sum_{n=0}^{M-2} r_n}{2}\right)^2\right]\right)^{D/2} + b(N,\delta,k) \quad \exists M \in \{1, \cdots, 2k-1\}$$

$$\leq c(N,\delta,k)\left((r_M^{\max})^2 - \left(\frac{d_{ij} - \sum_{n=0}^{M-2} r_n}{2}\right)^2\right)^{D(2k-1)/2} + b(N,\delta,k) \tag{42}$$

i. e. ,

$$d_{ij} \leq \sum_{n=0}^{M-2} r_n + 2\sqrt{(r_M^{\max})^2 - \left( \frac{\eta_{2k}(i,j) - b(N,\delta,k)}{c(N,\delta,k)} \right)^{\frac{2}{D(2k-1)}}}$$

$$\leq \sum_{n=0}^{M-2} r_n + 2\sqrt{(r_M^{\max})^2 - \left( \frac{\eta_{2k}(i,j)}{(N - \sqrt{-2N\ln\delta})^{2k-1}} - \frac{\sqrt{\frac{N\ln(1/2\delta)}{2}}}{N + \sqrt{-3N\ln\delta}} \right)^{\frac{2}{D(2k-1)}}} \qquad (43)$$

$$\leq \sum_{n=0}^{M-2} r_n + 2\sqrt{(r_M^{\max})^2 - \left( 1 - \frac{\sqrt{\frac{N\ln(1/2\delta)}{2}}}{N + \sqrt{-3N\ln\delta}} \right)^{\frac{2}{D(2k-1)}}}$$

$$\square$$

The motivation for introducing normalizedCN has been explained in detail in the main text. After introducing normalizedCN, we will derive Theorem 5.4, which we restate here as follows:

**Proposition D.7.** *(Latent space distance bound with $k$-hop CNs weighted by normalizedCN$(i,j)$). Originally, the contribution of each $k$-hop CN was assigned a value of 1. However, after introducing normalizedCN, the contribution of each $k$-hop CN is now given by $\sum_{c \in CN^k(i,j)} \frac{1}{|P_k(c)|}$. Therefore, we simply need to modify the overall contribution $\eta_{2k}(i,j)$ to $\frac{\eta_{2k}(i,j)}{\sum_{c \in CN^k(i,j)} 1/|P_k(c)|}$. For any $\delta > 0$, with probability at least $1 - \delta$, we have*

$$d_{ij} \leq \sum_{n=0}^{M-2} r_n + 2\sqrt{(r_M^{\max})^2 - \left( \left( \gamma \binom{\zeta}{2} \right)^{\frac{1}{D(k-1)}} \cdot \rho^N \right)^{\frac{2k-2}{2k-1}}} \qquad (44)$$

*where $\zeta$ is the maximum degree of all $k$-hop CNs of $(i,j)$ and $\rho \in [0,1]$ and $\gamma = \left( \frac{\eta_{2k}(i,j)}{(N - \sqrt{-2N\ln\delta})^{2k-1}} - \alpha \right)$. (The above only shows the case where the $k$-hop CNs are in symmetric positions, as in (9). The asymmetric case, as in (10), is very similar. )*

*Proof.* Consider the metric: the $k$-hop CNs we are interested in are also the $k$-hop CNs of several other pairs. We record the reciprocal of the sum of these pairs as the weight of the $k$-hop CNs.

$$P_k^n(i,j) \geqslant \left[ \prod_{p=1}^{k-1} A\left( r_p', \sum_{m=p+1}^{k} r_m, \sum_{m=p}^{k} r_m \right) \right]^n \cdot \left[ 1 - \prod_{p=1}^{k-1} A\left( r_p', \sum_{m=p+1}^{k} r_m, \left( D_i - \sum_{n=0}^{p-2} r_n \right)_+ \right) \right]^{N-n-1} \qquad (45)$$

$$P_k^n(i,j) \leqslant \left[ \prod_{p=1}^{k-1} A\left( r_p', \sum_{m=p+1}^{k} r_m, \left( D_i - \sum_{n=0}^{p-2} r_n \right)_+ \right) \right]^n \cdot \left[ 1 - \prod_{p=1}^{k-1} A\left( r_p', \sum_{m=p+1}^{k} r_m, \sum_{m=p}^{k} r_m \right) \right]^{N-n-1} \qquad (46)$$

$$
E\left[\eta_{\sum_{CNk}}(i)\right] \geqslant \binom{\zeta}{2} \left[\prod_{p=1}^{k-1} A\left(r'_p, \sum_{m=p+1}^{k} r_m, \sum_{m=p}^{k} r_m\right)\right]^n
$$

$$
\left[1 - \prod_{p=1}^{k-1} A\left(r'_p, \sum_{m=p+1}^{k} r_m, \left(D_i - \sum_{n=0}^{p-2} r_n\right)_+\right)\right]^{N-n-1}
$$

$$
\geqslant \binom{\zeta}{2} \left[\left(\frac{r'_{\min} - r_{\min}}{2}\right)^{D(k-1)}\right]^n \left[1 - \left((r_M^{\max})^2 - \left(\frac{D_i - \sum_{n=0}^{M-2} r_n}{2}\right)^2\right)^{\frac{D(k-1)}{2}}\right]^{N-n-1}
$$
(47)

$$
\eta_{2k}(i,j) \leq \frac{E\left[\eta_{2k}(i,j)\right]}{\binom{\zeta}{2}\left[\left(\frac{r'_{\min}-r_{\min}}{2}\right)^{D(k-1)}\right]^n \left[1 - \left((r_M^{\max})^2 - \left(\frac{D_i - \sum_{n=0}^{M-2} r_n}{2}\right)^2\right)^{\frac{D(k-1)}{2}}\right]^{N-n-1}} + \Delta^{2k-2}\sqrt{\frac{N\ln(1/2\delta)}{2}}
$$
(48)

Consider $\left(\rho^{D(k-1)}\right)^n \left(1 - \rho^{D(k-1)}\right)^{N-n-1} \leqslant (\max\{\rho, \xi\})^{D(k-1)N}$, where $\rho^{D(k-1)} + \xi^{D(k-1)} = 1^{D(k-1)}$. So We get :

$$
\eta_{2k}(i,j) \leq \frac{E\left[\eta_{2k}(i,j)\right]}{\binom{\zeta}{2}(\max\{\rho,\xi\})^{D(k-1)N}} + \Delta^{2k-2}\sqrt{\frac{N\ln(1/2\delta)}{2}}
$$
(49)

Note $\chi = \binom{\zeta}{2}(\max\{\rho,\xi\})^{D(k-1)N}$, we have: $c(N,\delta,k)' = \frac{c(N,\delta,k)}{\chi}$ and $b(N,\delta,k)' = b(N,\delta,k)$

$$
d_{ij} \leq \sum_{n=0}^{M-2} r_n + 2\sqrt{(r_M^{\max})^2 - \left(\frac{\eta_{2k}(i,j) - b(N,\delta,k)'}{c(N,\delta,k)'}\right)^{\frac{2}{D(2k-1)}}}
$$

$$
\leq \sum_{n=0}^{M-2} r_n + 2\sqrt{(r_M^{\max})^2 - \left(\frac{\eta_{2k}(i,j) - b(N,\delta,k)}{\frac{c(N,\delta,k)}{\chi}}\right)^{\frac{2}{D(2k-1)}}}
$$

$$
\leq \sum_{n=0}^{M-2} r_n + 2\sqrt{(r_M^{\max})^2 - \chi\left(1 - \frac{\sqrt{\frac{N\ln(1/2\delta)}{2}}}{N + \sqrt{-3N\ln\delta}}\right)^{\frac{2}{D(2k-1)}}}
$$
(50)

$$
\leq \sum_{n=0}^{M-2} r_n + 2\sqrt{(r_M^{\max})^2 - \left(\left(\gamma\binom{\zeta}{2}\right)^{\frac{1}{D(k-1)}} \cdot (\max\{\rho,\xi\})^N\right)^{\frac{2k-2}{2k-1}}}
$$

$$
= \sum_{n=0}^{M-2} r_n + 2\sqrt{(r_M^{\max})^2 - \left(\left(\gamma\binom{\zeta}{2}\right)^{\frac{1}{D(k-1)}} \cdot \rho^N\right)^{\frac{2k-2}{2k-1}}},
$$

where $\gamma = \left(\frac{\eta_{2k}(i,j)}{(N-\sqrt{-2N\ln\delta})^{2k-1}} - \alpha\right)$ and $\rho = \max\{\rho, \xi\}$.

$\square$

Additionally, in negative curvature spaces, the contribution of high-hop common neighbors becomes more significant. Euclidean space may underestimate it because the connectivity of nodes is more

dependent on local geometric structures, whereas in hyperbolic space, the global structure and curvature effects between nodes have a more pronounced impact on path propagation, meaning that the high-hop CNs information between nodes exhibits stronger structural dependence. In some tree-like graph structures, or in regions of graphs with negative curvature, a large number of high-hop CNs of various orders are more likely to occur. We hope that *normalizedCN* can assign a reasonable weight to each $k$-hop CNs in the hyperbolic space.

**Proposition D.8.** *When the latent space becomes a hyperbolic space with curvature $\kappa$,* normalizedCN *still remains effective, without the need to explicitly introduce $\kappa$ in the form of* normalizedCN.

*Proof.* The volume of a sphere or a ball in hyperbolic $n$-space with sectional curvature $\kappa$ is given by

$$V_\kappa(r) = c_{n-1} \int_0^r \left( \frac{\sinh(\sqrt{\kappa}t)}{\sqrt{\kappa}} \right)^{n-1} dt, \tag{51}$$

where $c_{n-1} := \frac{2\pi^{n/2}}{\Gamma(n/2)}$ is the $n-1$-dimensional area of a unit sphere in $\mathbb{R}^n$ (see [Chavel, 1995]).

Let $R = \frac{1}{\sqrt{-\kappa}}$, we have:

$$V_\kappa(r) = C_{n-1} R^{n-1} \int_0^r \sinh^{n-1} \frac{t}{R} dt \tag{52}$$

Using the recurrence formula for integrals involving hyperbolic functions $\int \sinh^n cx\, dx = \frac{1}{cn} \sinh^{n-1} cx \cosh cx - \frac{n-1}{n} \int \sinh^{n-2} cx\, dx \quad (n > 0)$, we finally obtain:

1. When $n$ is even ($n = 2k$):

$$A_n = f(n) + \sum_{k=1}^{\frac{n-2}{2}} (-1)^k \frac{(n-1)(n-3)\cdots[n-(2k-1)]}{n(n-2)(n-4)\cdots[n-(2k-2)]} f(n-2k) - (-1)^{\frac{n-2}{2}} \frac{(n-1)(n-3)\cdots 3 \cdot 1}{n(n-2)\cdots 4 \cdot 2} r \tag{53}$$

2. When $n$ is odd ($n = 2k+1$):

$$A_n = f(n) + \sum_{k=1}^{\frac{n-1}{2}} (-1)^k \frac{(n-1)(n-3)\cdots[n-(2k-1)]}{n(n-2)(n-4)\cdots[n-(2k-2)]} f(n-2k) \tag{54}$$

Where $f(n) = \frac{1}{an}(\sinh^{n-1} ar)(\cosh ar)$, $a = \sqrt{-\kappa}$, and $V_\kappa(r) = C_{n-1} R^{n-1} A_{n-1}$

It follows naturally that:

$$V(r) = V(1) \left( \frac{e^{ar} - e^{-ar}}{e^a - e^{-a}} \right)^D \cdot \left( \frac{e^{ar} + e^{-ar}}{e^a + e^{-a}} \right) \tag{55}$$

Clearly, when $\kappa$ approaches 0, which means the space degenerates from hyperbolic space to Euclidean space, we have:

$$\lim_{\kappa \to 0} \left( \frac{e^{ar} - e^{-ar}}{e^a - e^{-a}} \right)^D \cdot \left( \frac{e^{ar} + e^{-ar}}{e^a + e^{-a}} \right) = r^D \tag{56}$$

This is consistent with the result for Euclidean space.

We first need to find an upper bound for the volume of the intersection of two spheres in hyperbolic $n$-space with centers at $O_1$ and $O_2$ by scaling. In this case, we assume symmetry, as shown in Figure 7, where the upper bound for the volume of the intersection is the volume of a sphere with radius $\tau$. We can set up the following system of equations:

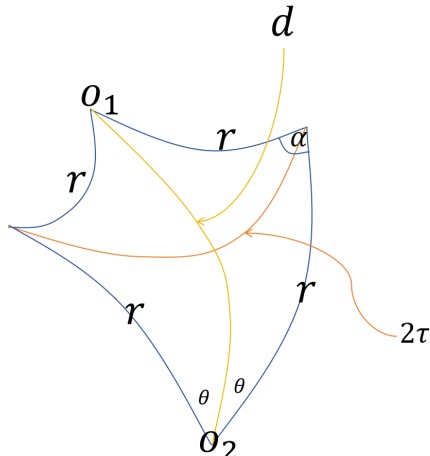

Figure 7: The upper bound for the volume of the intersection of two spheres in hyperbolic $n$-space with centers at $O_1$ and $O_2$ is the volume of a sphere with radius $\tau$.

$$\begin{cases} \cosh d = \cosh^2 r - \sinh^2 r \cdot \cos \alpha \\ \dfrac{\sinh d}{\sin \alpha} = \dfrac{\sinh r}{\sin \theta} \\ \cosh \tau = \cosh^2 r - \sinh^2 r \cdot \cos(2\theta) \end{cases} \tag{57}$$

Here, assuming the Gaussian curvature of the space is $\frac{1}{\pi^2}$, $d$ represents $\frac{d}{\pi}$, $r$ represents $\frac{r}{\pi}$, and $\tau$ represents $\frac{\tau}{\pi}$ for simplicity in form. The solution is:

$$\cosh \tau = \frac{8y^2(x-1)}{2x^2-1} - 1 \tag{58}$$

where $x = \cosh d$ and $y = \cosh r$.

Thus, when $D$ is greater than 64, we can almost assume that:

$$\frac{A\left(r_i, r_j, d_{ij}\right)}{V(1)} \leqslant \left(\frac{e^{ar} - e^{-ar}}{e^a - e^{-a}}\right)^D \cdot \left(\frac{e^{ar} + e^{-ar}}{e^a + e^{-a}}\right) \leqslant \left[(\sqrt{-\kappa})^r\right]^D \tag{59}$$

Similar to (42), we can derive:

$$\eta_{2k}(i, j) \leqslant c(N, \delta, k) \left(\sqrt{\kappa}^{\cosh^{-1}\left(\frac{8y^2(x-1)}{2x^2-1}-1\right)}\right)^{D(2k-1)} + b(N, \delta, k) \tag{60}$$

Since the range of $d_{ij}$ is $[0, 1]$, within this range, we can approximate $\frac{x-1}{2x^2-1}$ as $\frac{1}{5}(x-1)^{\frac{1}{2}}$, which is similar to the approach taken in Euclidean space. Ultimately, we can derive:

$$\cosh \frac{d_{ij}}{\kappa} \leqslant \left(\frac{5\left[\cosh\left(\log_{\sqrt{-\kappa}}\left(\frac{\eta-b}{c}\right)^{\frac{1}{D(2k-1)}}\right)+1\right]}{8\cosh^2 \frac{r_M^{min}}{\kappa}}\right)^2 + 1 \tag{61}$$

The form of (61) is very similar to (D). Following the same approach as in Euclidean space, after introducing normalizedCN, we can finally conclude that the effect of $d_{ij}$ brought by normalizedCN is consistent with that in Euclidean space.

Table 3: Statistics of dataset.

|  | **Cora** | **Citeseer** | **Pubmed** | **Collab** | **PPA** | **DDI** | **Citation2** |
|---|---|---|---|---|---|---|---|
| #Nodes | 2, 708 | 3, 327 | 18, 717 | 235, 868 | 576, 289 | 4, 267 | 2, 927, 963 |
| #Edges | 5, 278 | 4, 676 | 44, 327 | 1, 285, 465 | 30, 326, 273 | 1, 334, 889 | 30, 561, 187 |
| splits | random | random | random | fixed | fixed | fixed | fixed |
| average degree | 3.9 | 2.74 | 4.5 | 5.45 | 52.62 | 312.84 | 10.44 |

$\square$

# E    Proof of Theorem 6.1

*Proof.* When algorithm A can distinguish all the link pairs that algorithm B can distinguish, we consider algorithm A to be more expressive than algorithm B, provided there exists a pair of links that A can distinguish but B cannot. Therefore, we can prove this by constructing a simple counterexample.

Graph Autoencoder's prediction for link $(i, j)$ is $\langle \text{MPNN}(i, A, X), \text{MPNN}(j, A, X) \rangle$. So $\text{MPNN}(i, A, X) \odot \text{MPNN}(j, A, X)$ leads to GAE which is a part of OCN, so OCN can also express GAE. As MPNN can learn arbitrary functions of node degrees, OCN can express CN, RA, AA . we construct an example in Figure 4. White, green, orange, and yellow represent node features 0, 1, 2, and 3, respectively. $v_2$ and $v_3$ are symmetric, and GAE cannot distinguish $(v_1, v_2)$ and $(v_1, v_3)$. With node features ignored, $(v_1, v_2)$ and $(v_1, v_3)$ are symmetric, so CN, RA, AA, Neo-GNN, and BUDDY cannot distinguish them. NCN also degenerates into GAE, so it also cannot. However, $(v_1, v_2)$ and $(v_1, v_3)$ have different 2-hop CNs, which allows OCN to distinguish them.

$\square$

# F    Dataset Statistics

The statistics for each dataset are presented in Table 3. The data is randomly split with 70%, 10%, and 20% allocated to the training, validation, and test sets, respectively. Unlike the others, the collab dataset permits the use of validation edges as input for the test set.

Cora, Citeseer, and Pubmed are very small-scale graphs where link prediction is relatively easy. Existing models have nearly saturated performance on these small graphs. The real challenge in link prediction lies in large-scale graphs. Our model achieves significant performance improvements on these high-value, challenging large graphs, particularly on ogbl-ppa and ogbl-ddi.

# G    Model Architecture

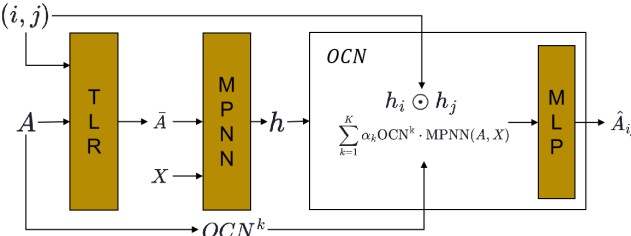

Figure 8: Architecture of OCN.

**Target Link Removal.**    We do not modify the input graph during the validation and test phases, where the target links remain hidden. For the training set, we remove the target links from $A$, and we define the modified graph as $\bar{A}$.

Table 4: Parameter configurations (OCN) across different datasets.

| Dataset | maskinput | mplayers | nnlayers | hiddim | ln | lnnn | res | jk | gnndp | xdp | tdp | gnnedp | predp | preedp | gnnlr | prelr | use_xlin | tallact |
|---------|-----------|----------|----------|--------|----|----|-----|----|-------|-----|-----|--------|-------|--------|-------|-------|----------|---------|
| Cora | T | 1 | 3 | 256 | T | T | F | T | 0.05 | 0.7 | 0.3 | 0.0 | 0.05 | 0.4 | 0.0043 | 0.0024 | T | T |
| Citeseer | T | 2 | 3 | 512 | F | T | T | F | 0.35 | 0.5 | 0.3 | 0.6 | 0.5 | 0.5 | 0.0005 | 0.0008 | T | F |
| Pubmed | T | 1 | 3 | 256 | T | T | F | T | 0.1 | 0.3 | 0.0 | 0.0 | 0.05 | 0.0 | 0.0097 | 0.002 | T | T |
| Collab | T | 1 | 3 | 256 | T | T | F | T | 0.05 | 0.7 | 0.3 | 0.0 | 0.05 | 0.4 | 0.0043 | 0.0024 | T | T |
| PPA | T | 1 | 3 | 64 | T | T | F | T | 0.0 | 0.0 | 0.0 | 0.0 | 0.0 | 0.0 | 0.0013 | 0.0013 | T | T |
| Citation2 | F | 5 | 3 | 32 | T | F | T | T | 0.28 | 0.5 | 0.3 | 0.2 | 0.1 | 0.12 | 0.00023 | 0.0009 | T | T |
| DDI | T | 3 | 3 | 64 | F | T | T | F | 0.25 | 0.13 | 0.38 | 0.5 | 0.10 | 0.13 | 0.00086 | 0.0008 | T | F |

**MPNN.** To generate the node representations $h$, we employ the MPNN framework. For each node $i$, the representation is obtained by:

$$h_i = \text{MPNN}(i, \bar{A}, X) \tag{62}$$

In the case of all target links, MPNN is executed only once.

**Predictor.** The link prediction task leverages node representations and the graph structure. The link representations for OCN are computed as follows:

$$z_{ij} = \text{MPNN}(i, A, X) \odot \text{MPNN}(j, A, X) + \sum_{k=1}^{K} \alpha_k \, \text{OCN}^k \cdot \text{MPNN}(A, X) \tag{63}$$

Here, $z_{ij}$ denotes the representation of the link $(i, j)$, The resulting representation is then processed to predict the likelihood of the link's existence:

$$\hat{A}_{ij} = \text{sigmoid}(\text{MLP}(z_{ij})) \tag{64}$$

## H  Experimental Settings

**Computing Setup.** We utilize PyTorch Geometric [Fey and Lenssen, 2019] and PyTorch [Paszke et al., 2019] for developing the models. All experiments are performed on a Linux server equipped with an Nvidia 4090 GPU.

**Baselines.** The results reported in [Wang et al., 2024] are directly used for comparison.

**Model Hyperparameters.** For hyperparameter tuning, we employ Optuna [Akiba et al., 2019] to conduct random search. The hyperparameters that yield the best validation scores are chosen for each model. The complete hyperparameter configuration is listed in Table 4 and Table 5.The key hyperparameters in our framework are defined as follows:

- `maskinput`: Boolean indicator for target link removal during training
- `mplayers`: Number of message passing layers in the GNN architecture
- `nnlayers`: Depth of Multilayer Perceptron (MLP) components
- `ln`: Layer normalization switch for MPNN modules
- `lnnn`: Layer normalization control for MLP components
- `jk`: Jumping Knowledge connection enablement
- `gnndp`: Dropout probability applied to GNN node representations
- `gnnedp`: Edge dropout ratio for graph adjacency matrices
- `predp`: Dropout rate in the prediction head network
- `preedp`: Edge dropout probability during prediction
- `gnnlr`: Learning rate for GNN parameter optimization
- `prelr`: Learning rate specific to predictor module

**Training Procedure.** We optimize models using the Adam optimizer. Results for all models are averaged from 10 runs with different random seeds.

**Computation Time.** The total computational cost for reproducing all experiments is shown Table 6.

Table 5: Parameter configurations (OCNP) across different datasets.

| Dataset | maskinput | mplayers | nnlayers | hiddim | ln | lnnn | res | jk | gnndp | xdp | tdp | gnnedp | predp | preedp | gnnlr | prelr | use_xlin | tallact |
|---|---|---|---|---|---|---|---|---|---|---|---|---|---|---|---|---|---|---|
| Cora | T | 1 | 3 | 256 | T | T | F | T | 0.05 | 0.7 | 0.3 | 0.0 | 0.05 | 0.4 | 0.0043 | 0.0024 | T | T |
| Citeseer | T | 3 | 1 | 64 | F | T | F | T | 0.12 | 0.73 | 0.88 | 0.07 | 0.19 | 0.06 | 0.0069 | 0.0010 | T | F |
| Pubmed | T | 1 | 3 | 256 | T | T | F | T | 0.1 | 0.3 | 0.0 | 0.0 | 0.05 | 0.0 | 0.0097 | 0.002 | T | T |
| Collab | T | 1 | 3 | 256 | T | T | F | T | 0.1 | 0.25 | 0.05 | 0.3 | 0.3 | 0.0 | 0.0082 | 0.0037 | T | T |
| PPA | T | 1 | 3 | 64 | T | T | F | T | 0.0 | 0.0 | 0.0 | 0.0 | 0.0 | 0.0 | 0.0013 | 0.0013 | T | T |
| Citation2 | F | 5 | 3 | 32 | T | F | T | T | 0.28 | 0.5 | 0.3 | 0.2 | 0.1 | 0.12 | 0.0002 | 0.0008 | T | T |
| DDI | T | 3 | 3 | 64 | F | T | T | F | 0.25 | 0.13 | 0.38 | 0.5 | 0.10 | 0.13 | 0.0009 | 0.0008 | T | F |

Table 6: Total time(s) needed in one run

|  | CORA | CITESEER | PUBMED | COLLAB | PPA | CITATION2 | DDI |
|---|---|---|---|---|---|---|---|
| **OCN** | 10 | 24 | 110 | 380 | 17010 | 18132 | 1600 |
| **OCNP** | 9 | 21 | 96 | 350 | 16770 | 21520 | 1131 |

# I Time and Space Complexity

Let $t$ represent the number of target links, $n$ the total number of nodes in the graph, and $d$ the maximum degree of a node. The time and space complexities of the existing models can be written as $O(B + Ct)$ and $O(D + Et)$, respectively. The constants $B$, $C$, $D$, and $E$ are independent of $t$, as summarized in Table 7. The derivation of complexity is as follows: models such as NCN [Wang et al., 2024], GAE [Kipf and Welling, 2016], and GNN, which utilize different structural features and operate on the original graph, exhibit similar complexities of $ndF + nF^2$. Specifically, the method by BUDDY [Chamberlain et al., 2023] uses a simplified version of MPNN, simplifying the complexity term $B$ to $ndF$. Additionally, Neo-GNN [Yun et al., 2021] requires precomputing the higher-order graph $A^l$, which results in time and space complexity of $O(nd^l)$. BUDDY hashes each node, resulting in $O(nh)$ time and $O(nh')$ space complexity. In contrast, SEAL [Zhang and Chen, 2018]'s $B$ is 0, as it does not run MPNN on the original graph. For each target link, a vanilla GNN simply requires feeding the feature vector to an MLP, yielding $C = F^2$. In addition to GAE's operation, BUDDY also hashes the structural features, which introduces a higher complexity per edge, $O(d^l)$, where $l$ is the number of hops Neo-GNN considers. For each target link, SEAL segregates a subgraph of size $O(d^{l'})$, where $l'$ represents the number of hops in the subgraph, and runs MPNN on it, which gives $C = d^{l'}F^2 + d^{l'+1}F$. NCN computes common neighbors in $O(d)$ time, pools the node embeddings with a complexity of $O(dF)$, and feeds them into an MLP, resulting in $O(F^2)$. NCNC-1 runs NCN for each possible common neighbor, leading to a time complexity of $O(d^2F + dF^2)$. For higher-order computations, NCNC-$K$ executes $O(d)$ times NCNC-$K$, resulting in a time complexity of $O(d^{K+1}F + d^K F^2)$. OCN computes $k$-hop CNs with complexity $O(d^k)$. The process of Schmidt orthogonalization has a time complexity of $O(k^2 n)$, and it computes (63) with a complexity of $O(dF)$. Finally, OCN feeds it into an MLP, resulting in $O(F^2)$. Similarly, the only difference between OCNP and OCN is that OCNP replaces the Schmidt orthogonalization process with Polynomial Filters, which results in a time complexity of $O(kn)$.

# J OCNP

According to Equation (1), we can express $CN_k$ and transform the Hadamard product into a Kronecker product($\otimes$) with many desirable properties. We have:

$$
\begin{aligned}
CN_k &= \bigcup_{2(k-1) < k_1 + k_2 \leq 2k, k_1 \leq k, k_2 \leq k} \left( P_1 A^{k_1} \right) \odot \left( P_2 A^{k_2} \right) \\
&= J \left( P_1 A^{k_1} \otimes P_2 A^{k_2} \right) K \\
&= J \left( P_1 \otimes P_2 \right) \left( A^{k_1} \otimes A^{k_2} \right) K \\
&\overset{k_1 = k_2 = k}{=} J \left( P_1 \otimes P_2 \right) (A \otimes A)^k K \\
&= J \left( P_1 \otimes P_2 \right) (W \otimes W)(\Sigma \otimes \Sigma)^k \left( W^\top \otimes W^\top \right) K,
\end{aligned}
\tag{65}
$$

Table 7: Scalability comparison. $h$, $h'$, $h''$: the complexity of hash function in BUDDY, where all $d \geq l$. $F$: the dimension of node representations. When predicting the $t$ target links, time and space complexity of existing models can be expressed as $O(B + Ct)$ and $O(D + Et)$, respectively.

| METHOD | B | C | D | E |
|---|---|---|---|---|
| GAE | $ndF + nF^2$ | $F^2$ | $nF$ | $F$ |
| NEO-GNN | $ndF + nF^2 + nd^l$ | $d^l + F^2$ | $nF + nd^l$ | $d^l + F$ |
| BUDDY | $ndF + nh$ | $h' + F^2$ | $nF + nh''$ | $F + h'$ |
| SEAL | $0$ | $d^{l'+1}F + d^{l'}F^2$ | $0$ | $d^{l'+1}F$ |
| NCN | $ndF + nF^2$ | $dF + F^2$ | $nF$ | $dF$ |
| NCNC | $ndF + nF^2$ | $d^2F + dF^2$ | $nF$ | $d^2F$ |
| OCN | $ndF + nF^2$ | $d^k + k^2n + dF + F^2$ | $nF$ | $dF$ |
| OCNP | $ndF + nF^2$ | $d^k + kn + dF + F^2$ | $nF$ | $dF$ |

where $P_1$ and $P_2$ are called selection matrices, defined as $P[j,k] = \delta(S[1,j],k)$, where $S \in \mathbb{R}^{2 \times h}$ describes the start and end points of each edge. Expanding $\left( \sum \otimes \sum \right)^k$, we abbreviate it as $[k]$:

$$[k] := \left( \sum \otimes \sum \right)^k = \begin{pmatrix} \lambda_1^k \lambda_1^k & & & & & & \\ & \lambda_1^k \lambda_2^k & & & & & \\ & & \ddots & & & & \\ & & & \lambda_1^k \lambda_h^k & & & \\ & & & & \lambda_2^k \lambda_1^k & & \\ & & & & & \ddots & \\ & & & & & & \lambda_h^k \lambda_h^k \end{pmatrix}, \quad (66)$$

Therefore, we can write $CN_k$ as $CN_{k_n} = U\,[k_n]\,V$. We note that the Frobenius inner product in the Schmidt orthogonalization process can be derived as follows:

$$
\begin{aligned}
\langle CN_{K_A}, CN_{K_B} \rangle_F &= \frac{1}{4} \left( \|CN_{K_A} + CN_{K_B}\|_F^2 - \|CN_{K_A} - CN_{K_B}\|_F^2 \right) \\
&= \frac{1}{4} \|U\,[k_A + k_B]\,V + U\,[k_A - k_B]\,V\| \, \|U\,[k_A + k_B]\,V - U\,[k_A - k_B]\,V\| \\
&= \sqrt{\sum_{i=1}^n \sigma_i^2 \left( U\,[k_A]\,V \right)} \cdot \sqrt{\sum_{i=1}^n \sigma_i^2 \left( U\,[K_B]\,V \right)},
\end{aligned}
$$
$$(67)$$

Here, $\sigma_i^2(A)$ represents the $i$-th singular value of $A$. Next, we abbreviate $\sqrt{\sum_{i=1}^n \sigma_i^2 (U\,[k_B]\,V)}$ as $\sqrt{\sum \sigma^2 (k_B)}$. Then, we can write the Schmidt orthogonalization process for $B$ as follows:

$$
\begin{aligned}
B - & \langle B, A_1 \rangle A_1 - \langle B, A_2 \rangle A_2 \cdots \cdots \\
&= U\,[k_B]\,V - \sqrt{\sum \sigma^2 (k_A)} \sqrt{\sum \sigma^2 (k_B)} U\,[k_1]\,V - \sqrt{\sum \sigma^2 (k_B)} \sqrt{\sum \sigma^2 (K_{A_i})} U\,[k_i]\,V \\
&= U \Omega V
\end{aligned}
$$
$$(68)$$

For a given term $\lambda_i^k \lambda_j^k$ in $[k]$, we abbreviate it as $\lambda_{ij}^k$. Therefore, after undergoing Schmidt orthogonalization, each eigenvalue in the diagonal matrix undergoes the following transformation:

$$\lambda_{ij}^{k_B} \rightarrow \sqrt{\Sigma\sigma^2\left(k_B\right)}\left(\sqrt{\frac{1}{\sum\sigma^2\left(k_B\right)}} - \sqrt{\sum\sigma^2\left(k_1\right)}\frac{\lambda_{ij}^{k1}}{\lambda_{ij}^{k_B}} - \sqrt{\Sigma\sigma^2\left(k_2\right)}\frac{\lambda_{ij}^{k_2}}{\lambda_{ij}^{k_B}}\cdots\cdot\right)\lambda_{ij}^{k_B}$$

$$= \left[1 - \left(\sum_{i=1}^{k_B-1}\sqrt{\sum\sigma^2\left(k_B\right)}\sqrt{\sum\sigma^2\left(k_B-i\right)}\lambda_{ij}^{-i}\right)\right]\lambda_{ij}^{k_B} \tag{69}$$

That is to say, the original $B$ is equivalent to the following transformation:

$$CN_{k_B} = U\begin{pmatrix}\ddots & & \\ & \lambda_{ij}^{k_B} & \\ & & \ddots\end{pmatrix}V \Rightarrow U\begin{pmatrix}\ddots & & \\ & \left[1 - \left(\sum_{i=1}^{k_B-1}\sqrt{\sum\sigma^2\left(k_B\right)}\sqrt{\sum\sigma^2\left(k_B-i\right)}\lambda_{ij}^{-i}\right)\right]\lambda_{ij}^{k_B} & \\ & & \ddots\end{pmatrix}V$$

$$= CN_{k_B}\begin{pmatrix}\ddots & & \\ & 1 - \left(\sum_{i=1}^{k_B-1}\sqrt{\sum\sigma^2\left(k_B\right)}\sqrt{\sum\sigma^2\left(k_B-i\right)}\lambda_{ij}^{-i}\right) & \\ & & \ddots\end{pmatrix} \tag{70}$$

We abbreviate $\begin{pmatrix}\ddots & & \\ & 1 - \left(\sum_{i=1}^{k_B-1}\sqrt{\sum\sigma^2\left(k_B\right)}\sqrt{\sum\sigma^2\left(k_B-i\right)}\lambda_{ij}^{-i}\right) & \\ & & \ddots\end{pmatrix}$ as $g(k_B)$. We then

select two different $CN_{k_A}$ and $CN_{k_B}$ to analyze the relationship between $g(k_A)$ and $g(k_B)$. When $k_B$ is not significantly larger than $k_A$, we note that there is:

$$\left[1 - \left(\sum_{i=1}^{k_A-1}\sqrt{\sum\sigma^2\left(k_A\right)}\sqrt{\sum\sigma^2\left(k_A-i\right)}\lambda_{ij}^{-i}\right)\right]\cdot\left[1 - \left(\sum_{i=1}^{k_B-1}\sqrt{\sum\sigma^2\left(k_B\right)}\sqrt{\sum\sigma^2\left(k_B-i\right)}\lambda_{ij}^{-i}\right)\right]$$

$$= \left[1 - \left(\sum_{i=1}^{k_A-1}\sqrt{\sum\sigma^2\left(k_A\right)}\sqrt{\sum\sigma^2\left(k_A-i\right)}\lambda_{ij}^{-i}\right)\right]^2$$

$$= \left[1 - \left(\sum_{i=1}^{k_A-1}\sqrt{\sum_{i=1}^{n}\sigma_i^2\left(U\left[k_A\right]V\right)}\sqrt{\sum_{i=1}^{n}\sigma_i^2\left(U\left[k_A-i\right]V\right)}\lambda_{ij}^{-i}\right)\right]^2 \tag{71}$$

We assume that the introduction of $J\left(P_1 \otimes P_2\right)$ and $K$ does not significantly alter the larger eigenvalues in the adjacency matrix $A$. Therefore, we consider the larger singular values to correspond to the larger eigenvalues. Assuming we only consider the top $h$ largest singular values, we have:

$$g(k_A) \odot g(k_B) = \sum_{\lambda}\left[1 - \left(\sum_{i=1}^{k_A-1}\sqrt{\sum_{i=1}^{n}\sigma_i^2\left(U\left[k_A\right]V\right)}\sqrt{\sum_{i=1}^{n}\sigma_i^2\left(U\left[k_A-i\right]V\right)}\lambda_{ij}^{-i}\right)\right]^2$$

$$\sim h - \sqrt{\frac{(h+1)^2\left(\lambda_1^2+\lambda_2^2+\cdots\lambda_4^2\right)}{\lambda_1^2+\lambda_1^4+\cdots+\lambda_n^2+\cdots}} \sim 0 \tag{72}$$

This means that in our framework, $g(k_A)$ and $g(k_B)$ are approximately orthogonal.

Thus, we can obtain OCNP through the following operations: Let the $k$-th term of the selected polynomials be $T_n$. Then, $OCN^k = CN^k \operatorname{diag}(T_k)$, and we only need to replace the step in Algorithm 2 with:

Table 8: Results on link prediction benchmarks. The format is average score $\pm$ standard deviation. +tricks means model with tricks of PLNLP.

|  | COLLAB | PPA | CITATION2 | DDI |
| --- | --- | --- | --- | --- |
| METRIC | HITS@50 | HITS@100 | MRR | HITS@20 |
| NODE2VEC | 41.36±0.69 | 27.83±2.02 | 53.47±0.12 | 21.95±1.58 |
| DEEPWALK | 50.37±0.34 | 28.88±1.53 | 84.48±0.30 | 26.42±6.10 |
| LINE | 55.13±1.35 | 26.03±2.55 | 82.33±0.52 | 10.15±1.69 |
| PLNLP | 70.59±0.29 | 32.38±2.58 | 84.92±0.29 | 90.88±3.13 |
| GIDN | 70.96±0.55 | - | - | - |
| OCN | 72.43±3.75 | 69.79±0.85 | 88.57±0.06 | 97.42±0.34 |
| OCNP | 67.74±0.16 | 74.87±0.94 | 87.06±0.27 | 97.65±0.38 |
| NCN+TRICKS | 68.04±0.42 | - | - | 90.83±2.83 |
| OCN+TRICKS | 69.03±0.94 | 69.23±0.39 | 88.97±0.12 | 94.25±0.71 |
| OCNP+TRICKS | 69.89±0.22 | 73.44±0.73 | 88.79±0.21 | 97.43±0.29 |

$$OCN^k = CN^k \operatorname{diag}(T_k). \tag{73}$$

## K    Comparison with other Link Prediction Models

A key strength of GNNs lies in their inherent capacity to preserve permutation equivariance, meaning that edges with identical structural patterns—referred to as isomorphic edges—can give the same prediction. On the other hand, traditional node embedding methods, such as Node2Vec [Grover and Leskovec, 2016], LINE [Tang et al., 2015], and DeepWalk [Perozzi et al., 2014], often provide inconsistent results for isomorphic edges, which can impair their ability to generalize. In our study, we compared the performance of our proposed methods against these well-established node embedding methods, using several OGB datasets. Furthermore, PLNLP [Wang et al., 2021] and GIDN [Wang et al., 2022] improve their performance by employing a variety of training strategies, such as adjustments to the loss function and data augmentation techniques. As seen in our experiments (Table 8), we also applied the PLNLP tricks. While these adjustments did not yield substantial improvements, our models still delivered superior performance compared to both the node embedding methods and the models incorporating training tricks, irrespective of whether the tricks were applied.

## L    Ablation of MPNN

We provide an ablation study on the MPNN used in OCN and OCNP. The results are shown in Table 9. The MPNN models include GIN [Xu et al., 2019], GraphSage [Hamilton et al., 2017], MPNN with MAX aggregation, MPNN with SUM aggregation, MPNN with MEAN aggregation, and GCN [Kipf and Welling, 2017].

## M    Scalability Comparison on Datasets

The time and memory consumption of models on different datasets are shown in Figure 9. On these datasets, we observe results that are somewhat similar to those on the ogbl-collab dataset in Figure 5. OCN and OCNP generally scale better than Neo-GNN. SEAL has the worst scalability. The memory overhead of OCN is comparable to or slightly higher than NCN. In general, both the time and memory overhead of OCNP are better than those of OCN.

Liang et al. [2024] experimentally emphasizes the importance of explicitly incorporating NCN (number of common neighbors)-dependent structural information and highlights that ordinary GNNs cannot learn such structural patterns. However, Dong et al. [2024] further demonstrates that while ordinary GNNs indeed cannot learn CN, they can acquire CN knowledge by introducing noise as auxiliary input (though the learned CN contains variance). This approach significantly increases both parameter size and computational overhead, resulting in poor scalability. Our model exhibits substantially better scalability and efficiency compared to MPLP [Dong et al., 2024].

Table 9: Ablation study on MPNN.

| Dataset | Model | GCN | GIN | GraphSage | MAX | SUM | MEAN |
|---|---|---|---|---|---|---|---|
| Cora | GAE | 89.01±1.32 | 70.45±1.88 | 70.59±1.70 | 61.63±4.43 | - | - |
| | NCN | 89.05±0.96 | 70.62±1.68 | 70.94±1.47 | 66.53±2.27 | - | - |
| | OCN | 89.82±0.91 | 73.55±1.91 | 53.14±1.87 | 46.94±1.84 | 45.46±1.32 | 45.29±1.70 |
| | OCNP | 90.06±1.01 | 75.09±1.02 | 74.96±1.25 | 67.92±1.75 | 71.78±1.42 | 71.90±1.37 |
| Citeseer | GAE | 91.78±0.94 | 61.21±1.18 | 61.23±1.28 | 53.02±3.75 | - | - |
| | NCN | 91.56±1.43 | 61.58±1.18 | 61.95±1.05 | 53.40±2.34 | - | - |
| | OCN | 89.57±1.97 | 66.29±4.09 | 65.87±3.73 | 88.91±3.27 | 90.05±2.28 | 93.62±1.30 |
| | OCNP | 89.95±2.34 | 68.69±3.26 | 69.95±4.86 | 89.13±2.82 | 90.52±1.66 | 93.41±1.02 |
| Pubmed | GAE | 78.81±1.64 | 59.00±0.31 | 57.20±1.37 | 55.08±1.43 | - | - |
| | NCN | 79.05±1.16 | 59.06±0.49 | 58.06±0.69 | 56.32±0.77 | - | - |
| | OCN | 83.96±0.51 | 65.10±1.14 | 63.80±1.06 | 52.43±6.07 | 61.82±0.58 | 52.62±6.01 |
| | OCNP | 82.32±1.21 | 62.39±1.65 | 60.48±1.20 | 56.90±2.55 | 59.48±2.49 | 54.43±2.27 |
| collab | GAE | 36.96±0.95 | 38.94±0.81 | 28.11±0.26 | 27.08±0.61 | - | - |
| | NCN | 64.76±0.87 | 64.38±0.06 | 63.94±0.43 | 64.19±0.18 | - | - |
| | OCN | 68.19±0.21 | 72.43±3.75 | 66.12±0.34 | 65.44±0.96 | 65.35±0.40 | 65.36±0.34 |
| | OCNP | 67.74±0.16 | 57.70± 3.56 | 58.26±2.64 | 59.31±1.26 | 60.30±3.08 | 59.09±1.11 |
| ppa | GAE | 19.49±0.75 | 18.20±0.45 | 11.79±1.02 | 20.86±0.81 | - | - |
| | NCN | 61.19±0.85 | 47.94±0.89 | 56.41±0.65 | 57.31±0.30 | - | - |
| | OCN | 69.79±0.85 | 67.29±0.91 | OOM | OOM | 59.23±0.56 | OOM |
| | OCNP | 74.87±0.94 | 73.08±1.03 | OOM | OOM | 69.59±1.05 | OOM |
| Citation2 | OCN | 88.57±0.06 | 70.09±2.77 | OOM | OOM | 0.24±0.01 | OOM |
| | OCNP | 87.06±0.27 | OOM | OOM | OOM | 0.34±0.09 | OOM |
| DDI | OCN | 97.42±0.34 | 55.88±1.71 | 49.33±13.66 | 0.45±0.33 | 0.68±0.28 | 12.27±4.43 |
| | OCNP | 97.65±0.38 | 52.38±2.98 | 49.03±12.72 | 0.19±0.15 | 1.04±0.46 | 6.94 ±2.71 |

# N   Discussion Highlighting Differences from NCN

The primary motivation of NCN (and NCNC) lies in proposing a new architecture - **MPNN-then-SF** - aiming to address various limitations of the previous two major architectures (SF-then-MPNN and SF-and-MPNN). Here, NCN is just one instantiation of MPNN-then-SF, while NCNC is merely an iterative version of NCN. Neither focuses on incorporating higher-order CN; in fact, NCN found that attempting to introduce higher-order CN leads to performance degradation, and causes out-of-memory (OOM) issues on large graphs.

In contrast, our work directly targets the utilization of higher-order CN from the outset. We systematically investigate and summarize why existing methods of incorporating higher-order CN perform poorly. Ultimately, we identified the two key phenomena mentioned in our study - these are precisely what cause the performance deterioration when introducing higher-order CN. Therefore, our core motivation is to unlock the potential of higher-order CN by addressing these two phenomena.

Equation (7) may appear similar to NCN at first glance. However, our primary motivation and contribution lie in employing two key methods to incorporate higher-order CN for addressing the two phenomena we first discovered. Thus, our core contribution focuses on the preprocessing of CN before it reaches Equation (7) - the equation itself merely represents a summation step. The structural resemblance to NCN exists because our innovation doesn't involve architectural modifications, hence we retained NCN's MPNN-then-SF framework. We also analyzed the performance using SF-and-MPNN structure in our ablation studies.

We focus on the values of: $\alpha_1$ for $OCN^1$ and $\alpha_2$ for $OCN^2$. To ensure fair comparison with OCN, we also introduced higher-order CN to NCN and analyzed: $\alpha_1$ for $CN^1$ and $\alpha_2$ for $CN^2$. We present the results from Figure 10a to Figure 10l.

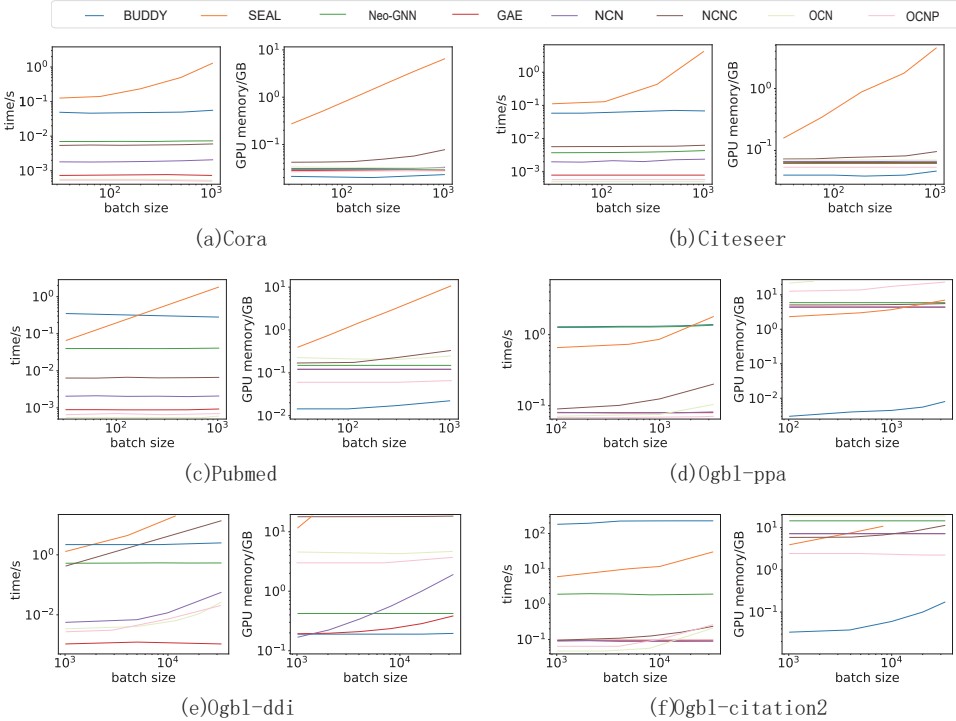

Figure 9: Inference time and GPU memory on datasets. The process we measure includes preprocessing, MPNN, and predicting one batch of test links.

Across all datasets, the learned $\alpha_1$ and $\alpha_2$ values are highly consistent between OCN and OCNP. Additionally, $\alpha_1$ and $\alpha_2$ remain relatively stable across all epochs (we only show epoch 1, epoch 50 and epoch 100 due to space limitations), with $\alpha_1$ always being greater than $\alpha_2$.

Through comparative analysis, we can draw the following conclusions:

1. The orthogonalization and normalization in our method lead to significantly faster convergence.

2. First-order neighbors are more important than second-order. Since NCN does not perform orthogonalization between CN1 and CN2, it results in a certain similarity between CN1 and CN2, which leads to instability in their importance. Therefore, NCN fails to maintain the inductive bias where $\alpha_2 < \alpha_1$. On large graphs, NCN suffers from OOM issues and cannot complete the linear combination of higher-order CN.

3. NCN reported performance degradation when incorporating higher-order CN which is likely caused by the two key phenomena discovered in our work.

## O   Analysis of the Performance Gap Resulting from Changing the Aggregation Strategy from Summation to Concatenation

We first formulate the final step of our original model OCN (OCNP) as:

$$\text{MLP}\left(\text{MPNN}(i, A, X) \odot \text{MPNN}(j, A, X) + \alpha_1 \, \text{OCN}^1 \cdot \text{MPNN}(A, X) + \alpha_2 \, \text{OCN}^2 \cdot \text{MPNN}(A, X)\right) \tag{74}$$

And the final step of OCN-CAT (OCNP-CAT) is formulated as:

$$\text{MLP}\left(\text{MPNN}(i, A, X) \odot \text{MPNN}(j, A, X) \,\|\, \alpha_1 \, \text{OCN}^1 \cdot \text{MPNN}(A, X) \,\|\, \alpha_2 \, \text{OCN}^2 \cdot \text{MPNN}(A, X)\right) \tag{75}$$

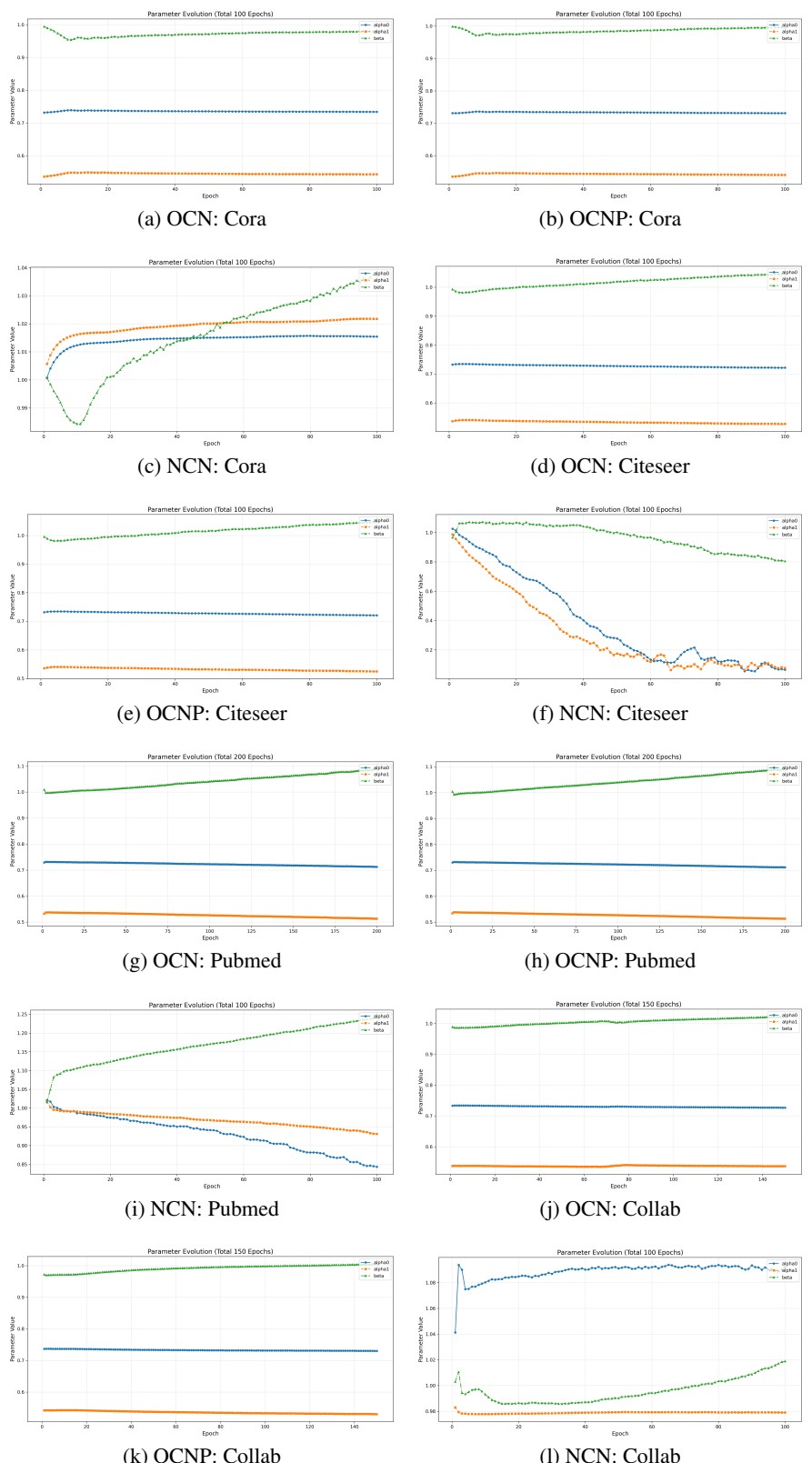

Figure 10: In the figure, $\alpha_0$, $\alpha_1$, and $\beta$ represent the learned coefficients for $OCN^1$, $OCN^2$, and $\text{MPNN}(i, A, X) \odot \text{MPNN}(j, A, X)$ respectively in the expression $\text{MPNN}(i, A, X) \odot \text{MPNN}(j, A, X) + \sum_{k=1}^{K} \alpha_k \, \text{OCN}^k \cdot \text{MPNN}(A, X)$.

**Algorithm 1** ORTHOGONALIZATIONOVERBATCH

---

**Input:** $\{CN_t^k\}_{k=1}^K$ over a mini-batch $\mathcal{B}_t = \{\{CN_t^k\}_{k=1}^K\}_t$;
Truncated polynomial order $K$;
Running inner product $\{\{\hat{\xi}^i\}_{i=1}^K\}_{t-1}$ over the last mini-batch.
**Output:** Orthogonalized data $\{OCN_t^k\}_{k=1}^K$
Initialize $OCN_t^1 \leftarrow CN_t^1/\|CN_t^1\|$
**for** $k = 2$ **to** $K$ **do**
  **if** training **then**
    **for** $i = 1$ **to** $k - 1$ **do**
      $\xi_t^i \leftarrow \langle CN_t^k, OCN_t^i \rangle$             ▷ Inner product within this batch
      $\beta_t \leftarrow 1/(t+1)$
      $\hat{\xi}_t^i \leftarrow (1 - \beta_t)\hat{\xi}_{t-1}^i + \beta_t\xi_t^i$      ▷ Maintain the global running inner product between mini-batches
    **end for**
  **end if**
  $CN_t^{k\perp} \leftarrow \sum_{i=1}^{k-1} \hat{\xi}^i \cdot OCN_t^i$
  $OCN_t^k \leftarrow CN_t^{k\perp}/\|CN_t^{k\perp}\|$
**end for**
**return** $\{OCN_t^k\}_{k=1}^K$

---

For the -CAT variant: 1. The MLP parameters are divided into three parts $(W_1\|W_2\|W_3)$. 2. The norms of different parts and $\alpha$ values jointly determine the importance of different orders.

Experimental results show the norms of the linear layers as presented from Figure 11a to Figure 11e, where $W_1$ corresponds to $\text{OCN}^1$ $W_2$ corresponds to $\text{OCN}^2$ and $W_3$ corresponds to $\text{MPNN}(i, A, X) \odot \text{MPNN}(j, A, X)$. For OCN (OCNP), the MLP parameters $W$ do not need to be split into $(W_1\|W_2\|W_3)$, and their norms are shown by the purple line (from Figure 11f to Figure 11j).

Our empirical analysis reveals two key insights regarding the model's learning behavior:

1. **Norm Distribution Pattern**: The parameter norms associated with higher-order neighbor representations exhibit monotonic growth across network layers (i.e., $\|W^{(k)}\|_2$ increases with order $k$), indicating the model's inherent preference for amplifying higher-order neighborhood information through geometric scaling.

2. **Operator Dynamics Comparison**: Through controlled experiments comparing concatenation (CAT) and summation (SUM) operators, we observe:

   - **Magnitude Disparity**: CAT implementations consistently produce larger parameter norms than their SUM counterparts ($\|W_{\text{CAT}}\|_2 > \|W_{\text{SUM}}\|_2$)

   - **Control Mechanism**: The SUM formulation $f_{\text{SUM}} = \text{lin}(\sum_{k=1}^K \alpha_k H^{(k)})$ enables explicit control over neighborhood order importance through learnable coefficients $\{\alpha_k\}_{k=1}^K$, where $\alpha_k$ directly determines the relative contribution of $k$-th order features.

   - **Coupling Effect**: In contrast, the CAT formulation $f_{\text{CAT}} = \text{lin}(\|_{k=1}^K \alpha_k H^{(k)})$ demonstrates parameter entanglement between projection matrices and coefficients, as the effective importance becomes jointly determined by $\alpha_k$ and the induced $\|W[:, d_k]\|_2$ norms in the linear transformation, where $d_k$ denotes the dimension slice for $k$-th order features.

# P Algorithm

Please refer to Algorithm 1 and Algorithm 2.

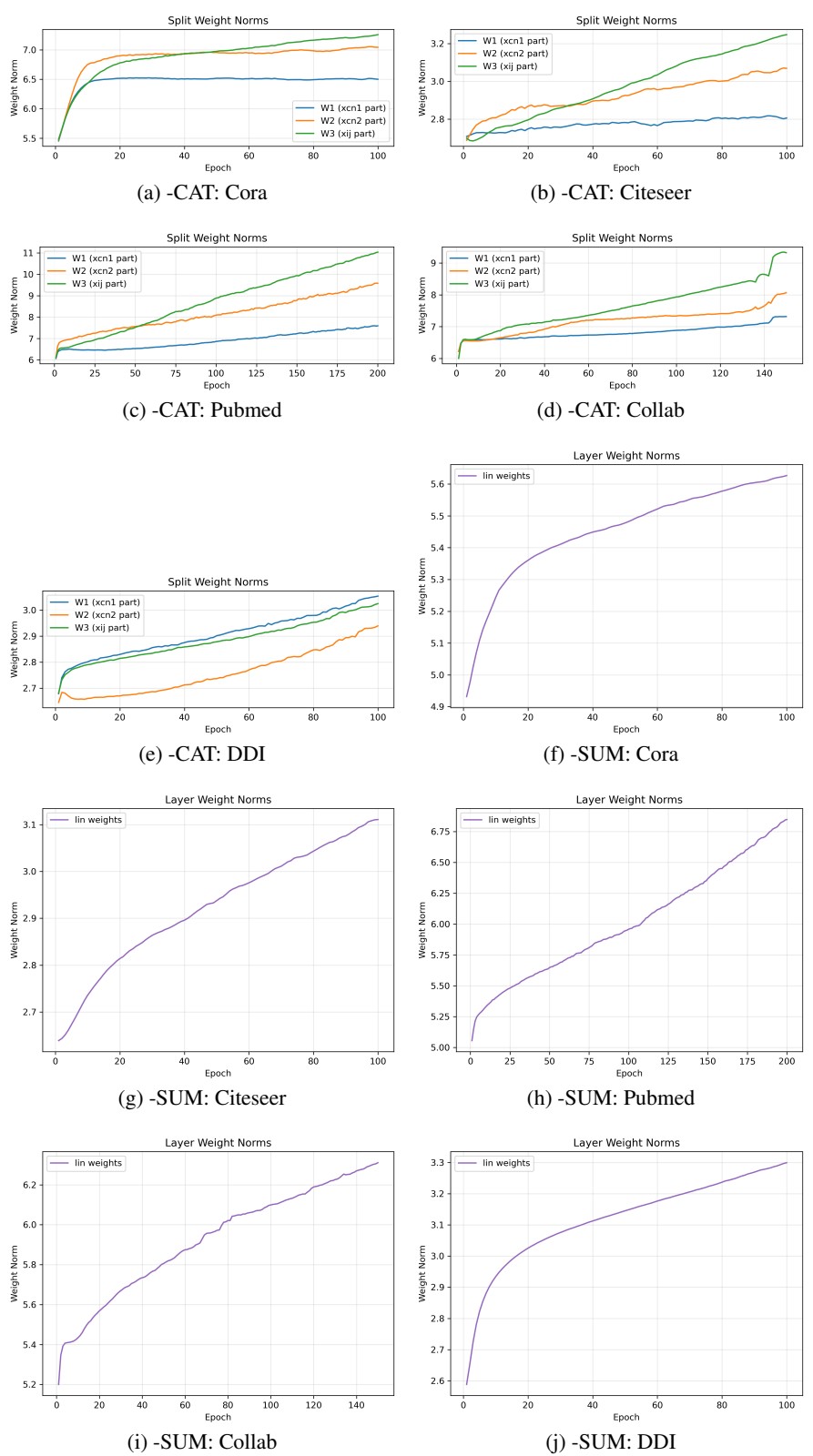

Figure 11: Norms of the linear layers where $W_1$ corresponds to $\text{OCN}^1$ $W_2$ corresponds to $\text{OCN}^2$ and $W_3$ corresponds to $\text{MPNN}(i, A, X) \odot \text{MPNN}(j, A, X)$. For OCN (OCNP), the MLP parameters $W$ do not need to be split into $(W_1 || W_2 || W_3)$, and their norms are shown by the purple line.

**Algorithm 2** OCNOVERBATCH

---

**Input:** $\{CN_t^k\}_{k=1}^K$ over a mini-batch $\mathcal{B}_t = \{\{CN_t^k\}_{k=1}^K\}_t$ ;Truncated polynomial order $K$;Running inner product $\{\{\hat{\xi}^i\}_{i=1}^k\}_{t-1}$ and $\hat{\psi}_{t-1}^k$ over the last mini-batch;input graph $A$, a node feature matrix $X$ and target links $\{(i_1, j_1), (i_2, j_2), \ldots, (i_t, j_t)\}$

**Learnable Parameters:** $\alpha_k$, all parameters in MPNNs

**Output:** link existence probability $\hat{A}_{ij}$

**for** $k = 1$ **to** $K$ **do**
  **if** training **then**
    $\psi_t^k \leftarrow CN^{k^\top} 1_h$
    $\gamma_t \leftarrow 1/(1+t)|$
    $\hat{\psi}_t^k \leftarrow (1 - \gamma_t)\hat{\psi}_{t-1}^k + \gamma_t \psi_t^k$  ▷ Convergence to the full graph of $CN^k \odot normalizedCN^k$
  **end if**
  $CN_t^k \leftarrow CN_t^k \cdot \text{diag}\left(\left(\hat{\psi}_t^k\right)^{-1}\right)$
**end for**

$$\{OCN_t^k\}_{k=1}^K \leftarrow \text{ORTHOGONALIZATIONOVERBATCH} \tag{76}$$

$$\left(\{CN_t^k\}_{k=1}^K, K, \{\{\hat{\xi}^i\}_{i=1}^k\}_{t-1}\right) \tag{77}$$

$Z_{ij} \leftarrow OCN(i, j, A, X)$ as described in (7)
$\hat{A}_{ij} \leftarrow \sigma(\text{MLP}(z_{ij}))$
**return** $\hat{A}_{ij}$

---

# Q   Theoretical Analysis with Barabási-Albert Model

The theoretical analysis is conducted on random graph models, which may not fully capture the structural properties of real-world networks. Therefore, we extend the theoretical arguments to the more realistic Barabási-Albert model. Given the numerous variants and extensions of this model, we select a class with broader universal significance. The specific construction method of a graph in this model is as follows:

**Definition Q.1.** *(Graph-Construction in Barabási-Albert) Form $G_n$ from $G_{n-1}$ by adding vertex $n$, sampling $m$ (with replacement) vertices $w_1, \ldots, w_m$ from $G_{n-1}$, and connecting $n$ to each $w_i$.*

*Conditioned on the past, the $w_i$ are i.i.d.: for $k < n$*

$$\Pr(w_i = k) = \frac{\deg_{n-1}(k)}{Z}, \qquad Z = \sum_{k=1}^{n-1} \deg_{n-1}(k). \tag{78}$$

**Definition Q.2.** *We define the score $s_{ij}$ based on the linking probability of $(i, j)$:*

$$Pr(i \sim j | s_{ij}, G_{max(i,j)-1}) = \frac{1}{1 + e^{\alpha(s_{ij} - r_{min(i,j)})}}, \tag{79}$$

*where $r$ satisfies $\deg(k) = NV(r_k) = NV(1)r_k^D$, $N = \#nodes$.*

**Proposition Q.3.** *When connecting an edge at vertex $b$, the expectation of degree of vertex $a$ (assuming $b > a$) is:*

$$E(\deg_b(a)) = m \frac{(2|a - b| - 1)!!}{2^{|a-b|}|a - b|!}. \tag{80}$$

*Simultaneously,*

$$E[Pr(a \sim b | G_{b-1})] = \frac{1}{4^{|a-b|}} \binom{2|a-b|}{|a-b|} \sim \frac{1}{\sqrt{\pi|a-b|}} \sim O(|a-b|^{-\frac{1}{2}}). \tag{81}$$

*Proof.* Consider that when forming $G_\xi$, the newly added vertex is $\xi$, and let the number of edges connecting $\xi$ to the previously existing vertices be $m$, i.e., $\deg_\xi(\xi) = m$. When forming $G_{\xi+1}$, the

newly added vertex is $\xi + 1$, and it is clear that the expected contribution of $\xi + 1$ to the degree of $\xi$ at this time is

$$\deg_{\xi+1}(\xi) = \frac{\deg_{\xi}(\xi)}{2m(\xi - 1)} \cdot m. \tag{82}$$

By analogy, when forming $G_{\xi+2}$, we have

$$\deg_{\xi+2}(\xi) = \frac{\deg_{\xi}(\xi) + \deg_{\xi+1}(\xi)}{2m\xi} \cdot m. \tag{83}$$

At this point, we can restate the problem as the following problem of finding the general term of a sequence:

The first term $E_1 = 0$, the second term $E_2 = m$, and for $v \geq 3$, the recurrence formula is:

$$E_v = \frac{E_1 + E_2 + \cdots + E_{v-1}}{2(v - 2)}. \tag{84}$$

According to the recurrence formula, for $v \geq 3$: $E_v = \frac{S_{v-1}}{2(v-2)}$.

Meanwhile, the partial sums satisfy: $S_v = S_{v-1} + E_v$. Substituting $E_v$:

$$S_v = S_{v-1} + \frac{S_{v-1}}{2(v - 2)} = S_{v-1} \cdot \frac{2v - 3}{2(v - 2)}. \tag{85}$$

Iterating from $v = 3$:

$$S_v = S_2 \prod_{j=3}^{v} \frac{2j - 3}{2(j - 2)} = m \prod_{j=3}^{v} \frac{2j - 3}{2(j - 2)}, \quad v \geq 2. \tag{86}$$

Splitting the product into two parts:

$$\prod_{j=3}^{v} \frac{2j - 3}{2(j - 2)} = \left( \prod_{j=3}^{v} \frac{1}{2} \right) \times \left( \prod_{j=3}^{v} \frac{2j - 3}{j - 2} \right). \tag{87}$$

The first part:

$$\prod_{j=3}^{v} \frac{1}{2} = \left( \frac{1}{2} \right)^{v-2}. \tag{88}$$

For the second part, let $k = j - 2$, then when $j = 3$, $k = 1$, and when $j = v$, $k = v - 2$:

$$\prod_{j=3}^{v} \frac{2j - 3}{j - 2} = \prod_{k=1}^{v-2} \frac{2(k + 2) - 3}{k} = \prod_{k=1}^{v-2} \frac{2k + 1}{k} \tag{89}$$

Now compute the product $\prod_{k=1}^{m} \frac{2k+1}{k}$, where $m = v - 2$:

$$\prod_{k=1}^{m} \frac{2k + 1}{k} = \frac{\prod_{k=1}^{m}(2k + 1)}{\prod_{k=1}^{m} k} = \frac{\prod_{k=1}^{m}(2k + 1)}{m!} \tag{90}$$

Given:

$$\prod_{k=1}^{m}(2k + 1) = \frac{(2m + 2)!}{2^{m+1}(m + 1)!} \tag{91}$$

Therefore:

$$\prod_{k=1}^{v-2} \frac{2k+1}{k} = \frac{(2(v-2)+2)!}{2^{(v-2)+1}((v-2)+1)!(v-2)!} = \frac{(2v-2)!}{2^{v-1}(v-1)!(v-2)!} \tag{92}$$

$$S_v = m \cdot \left(\frac{1}{2}\right)^{v-2} \cdot \frac{(2v-2)!}{2^{v-1}(v-1)!(v-2)!} = m \cdot 2^{-2v+3} \cdot \frac{(2v-2)!}{(v-1)!(v-2)!} \tag{93}$$

Note that:

$$\frac{(2v-2)!}{(v-1)!(v-2)!} = (v-1)\binom{2v-2}{v-1}, \tag{94}$$

Therefore:

$$S_v = m(v-1)\binom{2v-2}{v-1}2^{-2v+3}, \quad v \geq 2. \tag{95}$$

$$E_v = \frac{m(v-2)\binom{2v-4}{v-2}2^{-2v+5}}{2(v-2)} = m\binom{2v-4}{v-2}2^{-2v+4}. \tag{96}$$

That is:

$$E_v = m \cdot \frac{2^{v-2}(2(v-2)-1)!!}{(v-2)!} \cdot \left(\frac{1}{4}\right)^{v-2} = m \cdot \frac{(2v-5)!!}{(v-2)!} \cdot \frac{1}{2^{v-2}} \tag{97}$$

Returning to our original problem, we consider the expectation of the degree of vertex $a$ when connecting an edge at vertex $b$, and we can clearly obtain:

$$E(\deg_b(a)) = m\frac{(2|a-b|-1)!!}{2^{|a-b|}|a-b|!}. \tag{98}$$

And we have:

$$E[Pr(a \sim b|G_{b-1})] = \frac{1}{4^{|a-b|}}\binom{2|a-b|}{|a-b|} \sim \frac{1}{\sqrt{\pi|a-b|}} \sim O(|a-b|^{-\frac{1}{2}}). \tag{99}$$

$\square$

**Proposition Q.4.** *For any $\delta > 0$, with probability at least $\delta$, we have*

$$P_{2k}(i,j) \leq \frac{\prod^{2k} \frac{(2\Delta^++1)!!}{2^{\Delta^+}\Delta^+!} + \sqrt{\frac{\Delta^+\ln\delta^{-1}}{2}}}{2^{2k}(min(i,k_1,\cdots,k_{2k-1},j)-2)^{2k}}, \tag{100}$$

*where $\Delta^+ = max(|k_i - k_{i+1}|)$.*

*Proof.* The idea of the proof is very similar to our previous proof of Theorem 5.3. First, from the result of Theorem Q.3, we can easily obtain: For any $t > 0$,

$$\Pr\left(\deg_b(a) - \mathbb{E}[\deg_b(a)] \geq t\right) \leq \exp\left(-\frac{2t^2}{|a-b|m^2}\right). \tag{101}$$

Similar to the method used earlier to prove Theorem 5.3, we can similarly obtain:

$$P(a \sim b) \leq \frac{\frac{m(2|a-b|+1)!!}{2^{|a-b|}|a-b|!} + \sqrt{\frac{|a-b|\ln\delta^{-1}}{2}}}{2m(b-2)}, \tag{102}$$

Therefore we can obtain:

$$P_{2k}(i,j) = P(a \sim k_1 \sim k_2 \sim \cdots \sim b) \leq \frac{\prod^{2k} \frac{(2\Delta^+ + 1)!!}{2^{\Delta^+}\Delta^+!} + \sqrt{\frac{\Delta^+ \ln\delta^{-1}}{2}}}{2^{2k}(min(i,k_1,\cdots,k_{2k-1},j)-2)^{2k}}, \tag{103}$$

$\square$

**Proposition Q.5.** *For any $\delta > 0$, with probability at least $\delta$, we have*

$$s_{ij} \leq 2k\Big[\frac{1}{\alpha}\ln\big(\frac{2(N-2)}{\frac{(2N+1)!!}{2^N N!} + \frac{\sqrt{N\ln\delta^{-1}}}{4}} - 1\big) + \Big[\frac{1}{NV(1)}\big(m\frac{(2N+1)!!}{2^N N!} + \sqrt{\frac{Nm^2}{2}\ln\delta^{-1}}\big)\Big]^{\frac{1}{D}}\Big], \tag{104}$$

*where $N = \#nodes, k$ represents the order of the k-hop CNs.*

*Proof.* From the process of obtaining Equation (102) and Theorem Q.2, we can obtain:

$$\frac{1}{1 + e^{\alpha(d_{k_i k_{i+1}} - r_{min(k_i,k_{i+1})})}} \geq \frac{\frac{m(2|k_{i+1}-k_i|+1)!!}{2^{|k_{i+1}-k_i|}|k_{i+1}-k_i|!} + \sqrt{\frac{|k_{i+1}-k_i|\ln\delta^{-1}}{2}}}{2(max(k_{i+1},k_i)-2)}. \tag{105}$$

Let $|k_{i+1} - k_i| = t$, we can get:

$$d_{k_i k_{i+1}} \leq \frac{1}{\alpha}\ln\big(\frac{2((max(k_{i+1},k_i)-2)}{\frac{(2t+1)!!}{2^t t!} + \frac{\sqrt{t\ln\delta^{-1}}}{4}} - 1\big) + r_{min(k_i,k_{i+1})}. \tag{106}$$

So next we only need to compute $r_{min(k_i,k_{i+1})}$, let $k_- = min(k_i, k_{i+1})$.
We have

$$\deg(k_-) = NV(r_{k_-}) = NV(1)r_k^D \leq \frac{m(2|N-k_-|+1)!!}{2^{|N-k_-|}|N-k_-|!} + \sqrt{\frac{|N-k_-|m^2\ln\delta^{-1}}{2}}, \tag{107}$$

so:

$$r_{min(k_i,k_{i+1})} \leq \Big[\frac{1}{NV(1)}\big(\frac{m(2|N-k_-|+1)!!}{2^{|N-k_-|}|N-k_-|!} + \sqrt{\frac{|N-k_-|m^2\ln\delta^{-1}}{2}}\big)\Big]^{\frac{1}{D}} \tag{108}$$

In summary,

$$d_i j \leq 2k \cdot d_{k_i k_{i+1}} \leq 2k\Big[\frac{1}{\alpha}\ln\big(\frac{2((max(k_{i+1},k_i)-2)}{\frac{(2t+1)!!}{2^t t!} + \frac{\sqrt{t\ln\delta^{-1}}}{4}} - 1\big) + \Big[\frac{1}{NV(1)}\big(\frac{m(2|N-k_-|+1)!!}{2^{|N-k_-|}|N-k_-|!} + \sqrt{\frac{|N-k_-|m^2\ln\delta^{-1}}{2}}\big)\Big]^{\frac{1}{D}}\Big]. \tag{109}$$

$\square$

**Proposition Q.6.** *After introducing normalizedCN, for any $\delta > 0$, with probability at least $\delta$, we have*

$$s_{ij} \leq 2k\Big[\frac{1}{\alpha}\ln\big(\Big[-\frac{n-2}{N-n-1}W\big(-\frac{N-n-1}{n-2}C^{\frac{1}{n-2}}\big)\Big]^{-\frac{1}{k}} - 1\big) + \Big[\frac{1}{NV(1)}\big(m\frac{(2N+1)!!}{2^N N!} + \sqrt{\frac{Nm^2}{2}\ln\delta^{-1}}\big)\Big]^{\frac{1}{D}}\Big], \tag{110}$$

*where $W(\cdot)$ is **Lambert $W$ function**, $\zeta$ is the maximum degree of all k-hop CNs of $(i,j)$, the total number of paths of length $l$ between $i$ and $j$ is denoted as $\eta_l(i,j)$*

$$C = \frac{1}{\binom{\zeta}{2}} \frac{D^{2k-1}}{\eta_{2k} - D^{2k-2}\frac{\sqrt{N\ln\delta^{-1}}}{4}}, \tag{111}$$

*$D$ is the maximum degree on the graph.*

*Proof.* Using an idea very similar to the process of proving Theorem 5.4, we first obtain:

$$E\left[\eta_{\sum_{CNk}}(i)\right] \geqslant \binom{\zeta}{2}\left[\frac{\prod^{k}\frac{(2\Delta^{+}+1)!!}{2^{\Delta^{+}}\Delta^{+}!}+\sqrt{\frac{\Delta^{+}\ln\delta^{-1}}{2}}}{2^{k}(min(i,k_1,\cdots,k_{2k-1},j)-2)^{k}}\right]^{n}\left[1-\frac{\prod^{k}\frac{(2\Delta^{-}+1)!!}{2^{\Delta^{-}}\Delta^{-}!}+\sqrt{\frac{\Delta^{-}\ln\delta^{-1}}{2}}}{2^{k}(max(i,k_1,\cdots,k_{2k-1},j)-2)^{k}}\right]^{N-n-1}$$

(112)

Let $D$ be the maximum degree in the entire graph, then we have:

$$\eta_{\sum_{CNk}}(i) \leq \frac{D^{2k-1}\frac{\prod^{2k}\frac{(2\Delta^{+}+1)!!}{2^{\Delta^{+}}\Delta^{+}!}+\sqrt{\frac{\Delta^{+}\ln\delta^{-1}}{2}}}{2^{2k}(min(i,k_1,\cdots,k_{2k-1},j)-2)^{2k}}}{\binom{\zeta}{2}\left[\frac{\prod^{k}\frac{(2\Delta^{+}+1)!!}{2^{\Delta^{+}}\Delta^{+}!}+\sqrt{\frac{\Delta^{+}\ln\delta^{-1}}{2}}}{2^{k}(min(i,k_1,\cdots,k_{2k-1},j)-2)^{k}}\right]^{n}\left[1-\frac{\prod^{k}\frac{(2\Delta^{-}+1)!!}{2^{\Delta^{-}}\Delta^{-}!}+\sqrt{\frac{\Delta^{-}\ln\delta^{-1}}{2}}}{2^{k}(max(i,k_1,\cdots,k_{2k-1},j)-2)^{k}}\right]^{N-n-1}}+D^{2k-2}\frac{\sqrt{N\ln(2\delta)^{-1}}}{4}$$

(113)

let $P(\Delta^{+}) = \frac{\frac{(2\Delta^{+}+1)!!}{2^{\Delta^{+}}\Delta^{+}!}+\sqrt{\frac{\Delta^{+}\ln\delta^{-1}}{2}}}{2(min(i,k_1,\cdots,k_{2k-1},j)-2)}$, $P(\Delta^{-}) = \frac{\frac{(2\Delta^{-}+1)!!}{2^{\Delta^{-}}\Delta^{-}!}+\sqrt{\frac{\Delta^{-}\ln\delta^{-1}}{2}}}{2(min(i,k_1,\cdots,k_{2k-1},j)-2)}$ and $\prod^{k}P(\frac{\Delta^{+}+\Delta^{-}}{2}) = \lambda$, we have:

$$\eta_{\sum_{CNk}}(i) \leq \frac{D^{2k-1}}{\binom{\zeta}{2}\lambda^{n}(1-\lambda)^{N-n-1}} + D^{2k-2}\frac{\sqrt{N\ln(2\delta)^{-1}}}{4}.$$

(114)

We then get:

$$(\eta_{\sum_{CNk}}(i))^{n-2}(1-\eta_{\sum_{CNk}}(i))^{N-n-1} \leq \frac{1}{\binom{\zeta}{2}}\frac{D^{2k-1}}{\eta_{2k}-D^{2k-2}\frac{\sqrt{N\ln\delta^{-1}}}{4}}$$

(115)

So we can transform the problem into finding a closed-form upper bound for $x$ based on the inequality $x^{n-2}(1-x)^{N-n-1} \leq C$. We can solve this problem using the following method.

Clearly, we have:
$$x^{n-2}(1-x)^{N-n-1} \leq x^{n-2}e^{-(N-n-1)x}.$$

(116)

Let $k = n-2$, $m = N-n-1(>0)$, and replace the original equation with the **amplified upper bound**:

$$x^{k}e^{-mx} = c.$$

(117)

The left-hand side of it is always greater than or equal to the left-hand side of the original equation, so its solution $x_{up}$ must be less than or equal to the true solution of the original equation.

$$x^{k}e^{-mx} = c \implies k\ln x - mx = \ln c \implies x = e^{\frac{\ln c}{k}}e^{\frac{m}{k}x}.$$

(118)

$$xe^{-\frac{m}{k}x} = c^{1/k} \implies \left(-\frac{m}{k}x\right)e^{-\frac{m}{k}x} = -\frac{m}{k}c^{1/k}.$$

(119)

Let $y = -\frac{m}{k}x$, then $ye^{y} = -\frac{m}{k}c^{1/k}$.

Using the definition of the Lambert W function $W(z)e^{W(z)} = z$, we obtain $x_{up} = -\frac{k}{m}W\left(-\frac{m}{k}c^{1/k}\right)$ $(k = n-2, m = N-n-1)$.

$\square$

