# OpenReview forum: "OCN: Effectively Utilizing Higher-Order Common Neighbors for Better Link Prediction"
_NeurIPS.cc/2025/Conference — NeurIPS 2025 poster_

### Official Review · Reviewer_Ndeq · 2025-07-01

**Clarity:** 3
**Significance:** 2
**Originality:** 3
**Rating:** 4
**Confidence:** 3

**Summary:**

This paper focuses on the link prediction problem and proposes a higher-order common neighbor-based method by alleviating the issues of redundancy and over-smoothing. Experimental results demonstrate the effectiveness of the proposed method.

**Questions:**

Q1. According to Equation (1), the definition of k-hop common neighbors includes three cases: $k_1= k_2 = k$, $k_1 = k_2 + 1 = k$, and $k_1+1 = k_2 = k$. However, the meanings of these cases are not clearly explained in the main text and are instead relegated to the appendix.

Q2. The performance improvement of OCNP over OCN is not significant according to Table 6. This should be discussed in more details.

**Ethical Concerns:**

["NO or VERY MINOR ethics concerns only"]

**Final Justification:**

The authors addressed my concerns about k-hop common neighbors and the comparison with OCN. Thus, I will keep my positive score.

**Limitations:**

yes

**Paper Formatting Concerns:**

Figure 5 is too small and should be replotted.

**Quality:**

3

**Strengths And Weaknesses:**

S1. It is reasonable to use higher-order common neighbors to enhance the performance of link prediction.

S2. The problems of redundancy and over-smoothing are clearly described, and two techniques: coefficient orthogonalization and path-based normalization are proposed to address these problems.

S3. Theoretical analysis is provided to demonstrate the effectiveness of the proposed normalization technique on graphs.

W1. The definition of k-hop common neighbors is not clearly presented and is difficult to follow.

W2. The efficiency improvement achieved by OCNP is not significant.

---

> ### Author Rebuttal · Authors · 2025-07-31
>
> ### W1.
> Thank you deeply for your question! I hope the following answers will satisfy you.
> Specifically:
> $CN^k(u,v) = \sum_{2(k-1) < k_1 + k_2 \leq 2k, \atop k_1 \leq k, \ k_2 \leq k} \left(A^{k_1}\right)_u \odot \left(A^{k_2}\right)_v$
>
> The defined k-order CN is a scalar value, but it is defined for a specific node pair $(u, v)$. It represents: if a node has paths to $u$ of length $k₁$ and to $v$ of length $k₂$ (this refers to any path, not necessarily the shortest path), and the lengths $k₁$ and $k₂$ satisfy the given constraints, then this node is considered a k-order CN for the pair (u, v). The term $\left(A^{k_1}\right)_{u}$ (and similarly for $k_2$) actually counts the number of paths of length $k₁$ (or $k₂$) from this k-order CN node to $u$ (or $v$). This definition is also the commonly accepted definition of CN in the link prediction field, although some models define CN based on the Shortest Path Distance , whereas ours on the length of walks (paths).
> What we actually orthogonalize is a matrix of shape $(h, n)$, where $h$ is the number of edges in the current batch, and n is the total number of nodes. In plain language, the definition in Appendix B means: For each order $k$, we have three matrices of shape $(h, n)$. e.g. for 4-hop CN, the distance pairs can be (4,4); (3,4); or (4,3). For 7-hop CN, (7,7); (6,7); or (7,6). For such an $(h, n)$ matrix, suppose it corresponds to the distance pair (3,4). Then, the value $z$ at row $x$ and column $y$ indicates that for the edge $(u, v)$ corresponding to row $x$, the total number of paths from the node corresponding to column $y$ to $u$ that are exactly of length 3, and to $v$ that are exactly of length 4, is $z$. If the total number of paths is 0, it means the node corresponding to column $y$ cannot reach $u$ and $v$ via any paths of lengths exactly (3,4) respectively. Consequently, this node is not a 4-hop CN under the (3,4) criterion. However, it might be a 4-hop CN under the (4,4) or (4,3) criteria. If it satisfies none of these criteria, then it is not a 4-hop CN for the pair $(u, v)$.
>
> ### W2.
> Thank you for raising this point. The primary goal of OCNP is to discover a method that achieves mutually orthogonal results—even approximately—while bypassing the Gram-Schmidt orthogonalization process. Since Gram-Schmidt requires each subsequent matrix to operate on all preceding orthogonal bases, we aimed to directly derive orthogonalized results from each $(m,n)$-dimensional matrix in a single step.
>
> Through theoretical analysis, we proved that simply applying: $OCN^k = CN^k \cdot \text{diag}(T_k)$—i.e., right-multiplying the $(m,n)$ matrix by a diagonal matrix (with arbitrary polynomial basis elements on its diagonal)—yields results approximately equivalent to full Gram-Schmidt orthogonalization. This reduces the time complexity from $O(k^2)$ to $O(k)$ (see Table 7; theoretical analysis in Appendix I).
> Consequently, higher $K$ values deliver more pronounced asymptotic efficiency gains due to the $k^2 \to k$ complexity reduction. OCNP’s key contribution lies in providing a paradigm-shifting perspective for bypassing Gram-Schmidt orthogonalization in future work.
>
> ### Q1.
> Thank you for highlighting this point! Indeed, we have elaborated on the classification and analysis of both symmetric and asymmetric cases in the Appendix. The theoretical framework remains universally applicable to both scenarios, though the main text only shows the case where the $k$-hop CNs are in symmetric positions. The asymmetric case follows a highly similar analytical approach.
> We sincerely appreciate your suggestion and will clearly explain the meanings and implications of these cases in the main text in subsequent revisions.
>
> ### Q2.
> Thank you for your suggestion! Indeed, OCNP achieves virtually identical performance to OCN, as previously stated. The primary goal of OCNP is to discover a method that attains mutually orthogonal results—even approximately—while bypassing the Gram-Schmidt orthogonalization process. Though OCNP delivers approximately orthogonal outcomes, mathematically we can consider the final results of OCN and OCNP as approximately equivalent.
> The key contribution of OCNP lies in its potential to inspire future methodologies for circumventing Gram-Schmidt orthogonalization. We sincerely appreciate your insightful question!
>
> ### Figure 5 is too small and should be replotted.
>
> Thank you for your suggestion! We have redrawn the figure, but it cannot be displayed here due to constraints. The updated version will be incorporated in subsequent revisions.

---

> > ### Comment · Reviewer_Ndeq · 2025-08-07
> >
> > Thanks for your feedback! I think my concerns are addressed and I will keep my score.

---

> > > ### Author Response · Authors · 2025-08-08
> > >
> > > Thank you very much for your attention to our work. We are delighted that we could address your concerns! At the same time, we deeply appreciate your recognition!
> > >
> > > Best regards,
> > >
> > > Submission 6119 Authors

---

### Official Review · Reviewer_PbQ7 · 2025-07-02

**Clarity:** 1
**Significance:** 2
**Originality:** 2
**Rating:** 3
**Confidence:** 4

**Summary:**

A number of approaches have been proposed for link prediction in graphs. Recent ones includes Graph Neural Network based techniques. These techniques are often adopted to encode structural information by considering common neighbors between node pairs. One hop neighbors might not provide complete structural similarity indices in several graphs. Higher order neighbors reachable with $l$ hops is expected to provide more detailed information. Empirically, it has been observed that including higher order neighbors do not translate always to better performance in link prediction.

The present paper identifies to challenges in using higher order neighbors for link prediction. The first challenge is that of redundancy, meaning, as the order of neighbors considered grows, the nodes belonging to common neighbor set do not additionally include sufficient number of unique elements. The second challenge is the homogeneity of latent representations learned in a GNN across different nodes in the network. Both issues are definitely harmful for link prediction.

Authors propose two key ideas to alleviate these difficulties. Empirical results are presented to demonstrate that the correlation among representation of nodes in higher order neighborhood is very high. A orthogonalization is performed to remove these linear correlations. Thus a second level latent (linear?) representation is derived for the GNN based representation vectors. To allow scalable computation of the orthogonal components  the Gram-Schimdt batch orthogonalization is performed.

The second observation is that the nodes with high between-ness that are elements of multiple $l$-hop paths are over emphasized in the learned latent representation. And thus contribute adversely towards homogenization of the representations. This is addressed by simply normalizing the representation vectors with the count of the paths passing through these nodes.

Both the above approaches are nice and simple heuristics that is likely to improve performance of link prediction in a wide variety of network structures.

**Questions:**

1. Please define clearly the proposed method and the related orthogonalization/normalization techniques. Use clear mathematical equations of flow charts.

2. Highlight the novelty in terms of existing higher order neighbor based techniques.

4. Rewrite the caption of the Figures in the Results section for better interpretation.

**Ethical Concerns:**

["NO or VERY MINOR ethics concerns only"]

**Limitations:**

The presentation of the paper and its clarity should be improved. The originality and contributions in more broader terms should be highlighted.

**Paper Formatting Concerns:**

Many notations are undefined.

**Quality:**

2

**Strengths And Weaknesses:**

The paper proposes two effective approaches which reduces redundancy and homogeneity in GNN based link prediction using higher order common neighbor heuristics. Since using higher order common neighbors along with GNN appears to be a state of art in link prediction, the proposed method definitely is useful and effective.

The paper has extensive experimental results to demonstrate the effectiveness of the proposed techniques. Comparisons on benchmark data sets with various state of art methods are presented. The paper however lacks in novelty as all the methods used are well known in similar literature. The intuitions exploited in the work is also common knowledge in this research community and similar approaches were explored earlier.

The presentation of the paper is not very good. Several terms and concepts are undefined. For example, in Section 4 it is mention that "by analyzing the correlations coefficients among CNs of various orders...", although no definition of correlation coefficient or how it is computed is mentioned before. Similarly in Section 5 the exact steps/equations used in normalization is not clear.

There are a number of abbreviations that are undefined at first use and few inconsistent notations. legends of some of the plots like Figure 3,4, 5 are not easily understood.

The superiority of the proposed method cannot be established just on the basis o prediction accuracy. A combined analysis in terms of scalability and performance should be presented.

---

> ### Author Rebuttal · Authors · 2025-07-31
>
> We appreciate the reviewer for giving a very nice summary of the main merits and motivation of our paper, acknowledging that our paper proposes "**nice and simple heuristics** that is likely to improve performance of link prediction", "**two effective approaches** which reduces redundancy and homogeneity", and recognizing that "the proposed method definitely is **useful and effective**", "the paper has **extensive experimental results** to demonstrate the effectiveness". Below, we address your remaining concerns one by one.
> ### W1. methods used are well known
> While we leverages higher-order CNs, our core contribution and innovation are beyond merely introducing higher-order CNs to link prediction using existing techniques. The most important precondition is, **previous works exhibited low utilization efficiency and suboptimal effectiveness** when incorporating higher-order CNs, sometimes even hurting the performance. Our focus is thus **Effectively Utilizing Higher-Order CNs**.
>
> Neo-GNN computes features for high-order neighborhood overlap $N^{l_1}(i), N^{l_2}(j)$ as $\sum_{u \in N_1^{l_1}(i) \cap N_1^{l_2}(j)} A_{iu}^{l_1} A_{ju}^{l_2}f(d(u))$. But it simply concatenates structural features directly to the final representations, detaching them from MPNN, leading to reduced expressivity. And it only contains degree information and fundamentally fails to consider the features and influence of the high-order CNs themselves.
>
> BUDDY further considers overlap features and difference features: $\sum_{u\in N_{l_1}^{1}(i)\cap N_{l_2}^{1}(j)}1$, $\sum_{u\in N_{l}^{1}(i)-\cup_{l'=1}^{k}N_{l'}^{1}(j)}1$. But it only uses count information and simply concatenates structural features directly to the final representations.
>
> NCN is the 1st work that goes beyond merely using the count information of CN. Nevertheless, it does not utilize higher-order CNs nor consider the relationships between them. Prior to our work, the approach to incorporating higher-order CNs within the field remained solely at the level of computing count-based features and concatenating them. Beyond them, there have been no other successful works utilizing higher-order CNs.
>
> None have genuinely analyzed the higher-order CNs or explored how to truly unlock the potential of higher-order CNs. We are **the 1st to conduct an in-depth analysis of the reasons for failure** and, for the first time, successfully makes high-order CNs useful.
>
> ### W2. terms and concepts are undefined
> Thank you for your question. The correlation coefficients in Fig.1 used are just standard Pearson and the graph is Cora.
> The exact equations employed for normalization in Section 5 are detailed in Definition 5.1. Additionally, we provide intuitive explanations of their practical implications in lines 212-221. In our **response to Q1**, we provide a further, more formal and comprehensive explanation.
>
> All terms and concepts beyond common terminology are defined within the main text or appendices. We sincerely appreciate and hope your feedback.
>
> ### W3. abbreviations, notations inconsistent, and plots (Fig.3–5)
> Thanks! After a thorough verification, all abbreviations in the text were defined upon their first occurrence. Could you kindly provide more details regarding your concerns?
>
> Fig.3:
> Without path-based normalization, the upper bound of the probability for edge formation between (u, v) corresponds to the blue curve in Fig. 3. This is clearly suboptimal because increasing higher-order CNs information fails to tighten the probability bound. After introducing our normalization technique, the probability upper bound becomes the brown curve in Fig. 3. This provides a tighter and more reasonable probability estimation, consequently leading to significant performance improvement.
>
> **Fig. 4** is an example constructed in the proof of Theorem 6.1 (in Appendix E) to demonstrate that our model possesses superior expressive ability compared to other models. When algorithm A can distinguish all link pairs distinguishable by algorithm B, and there exists at least one link pair that A can distinguish but B cannot, we consider algorithm A to have stronger expressive ability than algorithm B. Fig. 4 represents one such counterexample. Nodes $v_2$ and $v_3$ are symmetric from $v_1$’s perspective, so GAE cannot distinguish between $(v_1, v_2)$ and $(v_1, v_3)$. If node features are ignored, $(v_1, v_2)$ and $(v_1, v_3)$ remain symmetric; thus CN, RA, AA, Neo-GNN, and BUDDY cannot distinguish them. This is because Neo-GNN and BUDDY only leverage the count of CNs without utilizing their feature information. Although NCN incorporates CNs' features, it only considers 1-hop CNs. In our example, differentiation between $(v_1, v_2)$ and $(v_1, v_3)$ occurs only when 2-hop CNs are considered—where differences in their features break symmetry. This enables OCN to distinguish the pairs, whereas NCN degenerates to GAE and consequently fails to differentiate them.
>
> **Fig. 5** illustrates the inference time versus batch size on Collab (left), and the GPU memory consumption versus batch size (right). Different colors represent distinct models. For a detailed theoretical analysis, please kindly refer to *Appendix I*. The complexity expressions can be decomposed into slope and constant terms.
>
>
> ### W4. combined analysis in terms of scalability and performance
> Thank you for your suggestion! Undoubtedly, our method excels in **both prediction accuracy and scalability**.
>
> Regarding prediction accuracy, according to the OGB leaderboard:
> **PPA**: Ours ranks 2nd. The top-ranked GraphGPT (SMTP) has a parameter count orders of magnitude larger than ours (and excels only on PPA and citation2). Refined-GAE is a purely empirical work relying on heavy hyperparameter search, and performs well only on PPA.
> **Collab**: Ours ranks 1st. HyperFusion is an A+B+C ensemble of existing methods (with much more parameters). Such hybrid models are prevalent among top leaderboard entries.
> **DDI**: Our model ranks 3rd, trailing the 2nd-place ELGNN by merely 0.12. The 1st-place is the ensemble method HyperFusion.
>
> In summary, across the OGB—excluding citation2 where performance gaps are minimal—our model achieves 1st, 2nd, and 3rd placements. It delivers superior performance with low parameter counts, beating many ensemble methods.
>
> Regarding scalability, we kindly direct the reviewer's attention to **Fig. 9 in our appendix**, where we have made a full comparison of inference time of the baselines. In summary,
> 1. OCN and NCN exhibit similar computational costs as they run the MPNN model only once.
> 2. In contrast, SEAL incurs substantially higher overhead, especially with larger batch sizes, since it re-executes MPNN per target link, causing sharp inference time growth.
> 3. OCN and OCNP generally scale better than Neo-GNN, while SEAL performs the worst.
>
> ### Q1: The complete algorithm for orthogonalization and path-based normalization is presented in Algorithm 2 on page 34.
> Specifically:  $CN^k(u,v) = \sum_{2(k-1) < k_1 + k_2 \leq 2k, \atop k_1 \leq k, \ k_2 \leq k} \left(A^{k_1}\right)_u \odot \left(A^{k_2}\right)_v.$
>
> The defined k-order CN is a scalar value, but it is defined for a specific node pair $(u, v)$. The term $\left(A^{k_1}\right)_{u}$ (and similarly for $k_2$) actually counts the number of paths of length $k₁$ (or $k₂$) from this k-order CN node to $u$ (or $v$).
> What we actually orthogonalize is a matrix of shape $(h, n)$, where $h$ is the number of edges, and n is the total number of nodes.
>
> Orthogonalization:
> The orthogonalization process is Gram-Schmidt, but applied to matrices. Orthogonality between matrices is defined by the Frobenius norm:$\|A^T B\|_F = 0 \quad \text{equivalent to} \text{tr}(A^T B) = 0$
> Section 4.1 addresses how to obtain the true inner product on a large real-world graph. In real graphs, the number of edges $m$ and nodes $n$ is extremely large. Performing the full matrix multiplication across the entire graph is computationally expensive. And training is performed batch-wise, so we only have access to a submatrix $(m', n)$ constructed from the edges within the current batch, where $m'$ is the number of edges in this batch.
> Our goal is therefore to approximate the inner product computed over all edges (i.e., the true inner product achieving orthogonality on the full graph) using only the edges visible within the current batch. We maintain a running inner product across batches, which accumulates the contribution of historical inner product information then combined with the inner product calculated from the current limited information via a weighted sum. As the number of training steps tends towards infinity, this running inner product converges to the full-graph inner product. In our experiments, we observed that the inner products indeed converged to a stable value.
>
> 4.2 aims to find a method that achieves mutually orthogonal results—even if approximately—while bypassing the Gram-Schmidt. We desire a way to directly compute the orthogonalized result from each $(m, n)$ matrix in a single step. Through proof, we discovered that simply applying $OCN^k = CN^k \cdot \text{diag}(T_k)$—that is, right-multiplying the $(m, n)$ matrix by a diagonal matrix (with an arbitrarily chosen polynomial basis placed on its diagonal). This reduces the time complexity from $O(k^2)$ to $O(k)$ (see Table 7).
>
> Path-based Normalization:
> The exact steps and equations used for normalization in Section 5 are detailed in Definition 5.1. We provide accessible descriptions of the underlying concepts and their practical implications in lines 212-221. The precise, step-by-step computational workflow for path-based normalization is presented within the first for loop of Algorithm 2 on page 34.
>
> Thank you once again for your questions and suggestions.
>
> ### Q2: We have addressed this concern in our response to W1. Thank you!
> ### Q3: We will revise the captions for better interpretation in the subsequent version.

---

> > ### Comment · Reviewer_PbQ7 · 2025-08-02
> >
> > Thank you for the clarification. Many of my concerns are addressed. The precise contribution which enhances the performance is now evident. I will re-look at the ratings.

---

> > > ### Author Response · Authors · 2025-08-07
> > >
> > > We would like to express our sincere gratitude for the valuable feedback provided by you and look forward to hearing whether our rebuttal has sufficiently addressed your concerns, or if you are willing to revise your scores. Once again, we sincerely appreciate your valuable suggestions and wish you all the best in your work.
> > >
> > > Best regards,
> > >
> > > Submission 6119 Authors

---

> > > ### Author Response · Authors · 2025-08-08
> > >
> > > Dear Reviewers,
> > >
> > > We sincerely appreciate your recognition of our work! Since many of your concerns have been addressed, we genuinely hope you could consider raising your score.
> > >
> > > Best regards,
> > > Submission 6119 Authors

---

### Official Review · Reviewer_8Aox · 2025-07-03

**Clarity:** 2
**Significance:** 2
**Originality:** 3
**Rating:** 4
**Confidence:** 3

**Summary:**

The paper introduces Orthogonal Common Neighbor (OCN), a novel method for link prediction that addresses the limitations of traditional Common Neighbors (CNs) and their higher-order variants. Existing approaches suffer from redundancy across different CN orders and over-smoothing, which hinder their effectiveness. OCN tackles these issues through two key techniques: orthogonalization, to remove redundancy between different-order CNs, and normalization, to alleviate over-smoothing effects. The proposed method achieves an average improvement of 7.7% over strong baselines on standard link prediction benchmarks. The authors provide both theoretical analysis and ablation studies to support the effectiveness of their approach.

**Questions:**

Q1. Have you considered using the proposed normalized Common Neighbors (CN) as an additional baseline, alongside standard heuristics such as CN, Adamic-Adar (AA), and Resource Allocation (RA)? It would be helpful to understand whether you experimented with this and how it compares in performance.

Q2. The formulation in Equation (4) is not entirely clear to me. Specifically, could you clarify why the function MPNN(A, X) is used to compute the score for a node pair (i, j)? Some additional explanation would help to better understand the underlying intuition.

Q3. Could you elaborate on the rationale for choosing different evaluation metrics across datasets (Hits@100 for some, Hits@20 for others, and MRR elsewhere)? This choice appears inconsistent and also deviates from recent works such as [n1]. A clarification on this point would strengthen the experimental section.

**Ethical Concerns:**

["NO or VERY MINOR ethics concerns only"]

**Final Justification:**

The authors addressed most of my initial concerns, including clarifications, additional experimental comparisons with further methods, and an extension of their theoretical analysis to the Barabási–Albert graph model (noting that, due to space constraints, only results are provided, not full proofs). These additions improve the completeness of the work, and I have accordingly raised my score to Borderline Accept.

However, my overall assessment is not fully positive. A major issue that remains unresolved is the clarity and readability of the paper—an aspect also noted by other reviewers. This significantly affects the accessibility and impact of the contribution. I hope the authors will take this feedback seriously and work to improve the exposition if the paper is accepted.

**Limitations:**

Yes

**Paper Formatting Concerns:**

No concerns.

**Quality:**

3

**Strengths And Weaknesses:**

**Strengths:**

S1. The problem addressed in this paper is well-motivated and clearly presented in the introduction, highlighting its relevance to the link prediction literature.

S2. The proposed method demonstrates strong empirical performance, consistently outperforming the considered competing methods.

S3. The paper includes a theoretical analysis that offers valuable insights into the behavior of the proposed approach.

**Weaknesses:**

W1. Some sections of the paper, particularly those describing the proposed method, would benefit from greater clarity. In particular, the discussion of the orthogonal scalability issues and the proposed solution in Section 4.1 is difficult to follow, and Section 4.2 is also not clearly explained. Improving the exposition of these key methodological components would enhance the overall readability and understanding of the paper.

W2. The ablation study suggests that higher-order common neighbors may introduce redundant information, potentially affecting the model’s stability and generalization ability (lines 286–288). This is concerning, as the use of higher-order neighbors is one of the central contributions of the paper. Evidence that performance may degrade beyond a certain number of hops raises questions about the robustness and general applicability of the proposed approach.

W3. The theoretical analysis is conducted on random graph models, which may not fully capture the structural properties of real-world networks. It would strengthen the paper to extend the theoretical justification to more realistic graph models such as Watts-Strogatz, Barabási-Albert, or Stochastic Block Models. If such an extension is not feasible, a discussion of this limitation should be included.

W4. The set of evaluation metrics is somewhat limited. For a more comprehensive assessment of predictive performance, additional metrics such as MRR (Mean Reciprocal Rank) should be reported on datasets like Cora, Citeseer, and PubMed, following the practice of recent works such as [n1].

W5. The experimental evaluation lacks some relevant and recent baselines, including [n1] and [n2], which also leverage heuristics and have demonstrated state-of-the-art performance in link prediction. Including these baselines would provide a more complete and convincing empirical comparison.

[n1] Mixture of Link Predictors on Graphs. NeurIPS 2024.

[n2] LPFormer: An Adaptive Graph Transformer for Link Prediction. KDD 2024.

**Overall recommendation**

The paper addresses an important problem in link prediction and proposes an interesting approach that combines higher-order common neighbors with orthogonalization and normalization techniques. The work shows good empirical results and includes some theoretical analysis, contributing to the significance of the topic.

However, the clarity of key methodological sections (W1) is lacking, making it difficult to fully grasp the proposed solution. The quality of the theoretical justification is limited by the use of overly simplistic graph models (W3), and the experimental evaluation would benefit from stronger baselines and more comprehensive metrics (W4, W5).

---

> ### Author Rebuttal · Authors · 2025-07-31
>
> ### W1.
> Thank you! Section 4.1 addresses how to obtain the true inner product on a large real-world graph. In real graphs the number of edges $m$ and nodes $n$ is extremely large. Performing the full (m, n) matrix multiplication across the entire graph is computationally expensive. And training is performed batch-wise, so we only have access to a submatrix $(m', n)$, where $m'$ is #edges in this batch.
> Our goal is to approximate the inner product computed over all edges (i.e., the true inner product achieving orthogonality on the full graph) using only the edges visible within the current batch. We maintain a running inner product across batches, which accumulates the contribution of historical inner product information and combined with the inner product calculated from the current batch. As $t$ approaches infinity, it converges to the full-graph inner product. In experiments, we observed the inner products indeed converged to a stable value.
>
> 4.2 aims to find a method that achieves mutually orthogonal results bypassing the Gram-Schmidt which has a high complexity. We desire a way to directly compute the orthogonalized result from each $(m, n)$ matrix in a single step. We discovered that applying $OCN^k = CN^k \cdot \text{diag}(T_k)$—that is, right-multiplying the $(m, n)$ matrix by a diagonal matrix (with an arbitrarily chosen polynomial basis placed on its diagonal)—sufficiently approximates the result originally requiring Gram-Schmidt. It reduces the time complexity from $O(k^2)$ to $O(k)$ (see Table 7).
>
> Due to space limit, we moved many details into Appendix B, C, I and J. We will make these sections more clear in the revised version.
>
> ### W2.
> Thank you! The previously suboptimal performance of OCN-3 (Table 2) stemmed from using the same hyperparameters as those for OCN, which failed to fully exploit the potential of higher-order CN. After further tuning the model using 3-hop CN with Optuna, we obtained new good results. As shown in the below table, the tuned OCN-3 model indeed outperforms the original model on certain datasets. But introducing higher-order CNs inevitably leads to a sharply increased computational burden and GPU memory consumption. We believe higher-order CNs can yield benefits, though this is also dependent on the specific model architecture and dataset characteristics. So we consider this work highly enlightening for subsequent research on how to efficiently utilize higher-order CNs.
>
> |Method|Cora(HR@100)|Citeseer(HR@100)|Pubmed(HR@100)|Collab(HR@50)|PPA(HR@100)|Citation2(MRR)|DDI(HR@20)|
> |-|-|-|-|-|-|-|-|
> |OCN|89.82±0.91|93.62±1.30|83.96±0.51|72.43±3.75|69.79±0.85|88.57±0.06|97.42±0.34|
> |OCN-3|90.96±0.64|92.33±0.85|87.12±1.40|73.04±2.13|OOM|87.34±0.38|90.06±1.29|
>
> ### W3.
> Thank you! We have extended the theoretical analysis to the Barabási-Albert model (given the numerous variants and generalized models, we selected one class demonstrating broad applicability), and we are privileged to complete the proof during rebuttal. Due to space constraints, we present only the conclusions here. Subsequently, we will provide the full detailed theoretical analysis and proofs in the camera ready version.
>
> **Construction.** Form $G_n$ from $G_{n-1}$ by adding vertex $n$, sampling $m$ (with replacement) vertices $w_1,\dots,w_m$ from $G_{n-1}$, and connecting $n$ to each $w_i$.
>
> Conditioned on the past, the $w_i$ are i.i.d.:
> for $k<n$
> $$
> \Pr(w_i=k)=\frac{\deg_{n-1}(k)}{Z},\qquad
> Z=\sum_{k=1}^{n-1}\deg_{n-1}(k).
> $$
>
> **Definition 1**
> We define the score based on the linking probability of $(i,j)$ $s_{ij}$:
> $$ Pr(i\sim j|s_{ij},G_{max(i,j)-1})=\frac{1}{1+e^{\alpha(s_{ij}-r_{min(i,j)})}},$$
>
> where $r$ satisfies $\deg(k)=NV(r_k)=NV(1)r^{D}_k$，$N = \text{number of nodes}.$
>
> **Proposition 1**
> When connecting an edge at vertex $b$, the expectation of degree of vertex $a$ (assuming $b>a$) is:
> $$E(\deg_b(a))=m\frac{(2|a-b|+1)!!}{2^{|a-b|} |a-b|!}.$$
> Simultaneously,
> $$ E[Pr(a\sim b|G_{b-1})]=\frac{1}{4^{|a-b|}}\tbinom{2|a-b|}{|a-b|}\sim \frac{1}{\sqrt{\pi |a-b|}} \sim O(|a-b|^{-\frac{1}{2}}).$$
>
>
> **Proposition 2**
> For any δ > 0, with probability at least δ, we have
>
> $$P_{2k}(i,j)\leq \frac{\prod^{2k}\frac{(2\Delta^++1)!!}{2^{\Delta^+} \Delta^+!}+\sqrt{\frac{\Delta^+ \ln \delta^{-1}}{2}}}{2^{2k}(min(i,k_1,\cdots,k_{2k-1},j)-2)^{2k}},$$
>
> where $\Delta=max(|k_i-k_{i+1}|).$
>
> **Proposition 3**
> For any δ > 0, with probability at least δ, we have
> $$s_{ij}\leq 2k\Bigl[\frac{1}{\alpha}\ln(\frac{2(N-2)}
>      { \frac{(2N+1)!!}{2^{N}N!}+
>                 \frac{\sqrt{N\ln\delta^{-1}}}{4}}-1)
>                 +\Bigl[\frac{1}{NV(1)}(m\frac{(2N+1)!!}{2^N N!}+\sqrt{\frac{Nm^2}{2}\ln{\delta^{-1}}})\Bigr]^{-\frac{1}{D}}\Bigr],$$
>
> where $N=\text{number of nodes}$, $k$ represents the order of the k-hop CNs.
>
> **Proposition 4**
> After introducing normalizedCN, for any δ > 0, with probability at least δ, we have
>
> $$s_{ij}\leq 2k\Bigl[\frac{1}{\alpha}\ln(\Bigl[
>   -\frac{n-2}{N-n-1}
>   W\!\Bigl(
>        -\frac{N-n-1}{n-2}C^{\tfrac{1}{n-2}}
>     \Bigr)
> \Bigr]^{-\tfrac{1}{k}}-1)
>                 +\Bigl[\frac{1}{NV(1)}(m\frac{(2N+1)!!}{2^N N!}+\sqrt{\frac{Nm^2}{2}\ln{\delta^{-1}}})\Bigr]^{-\frac{1}{D}}\Bigr],$$
> where $W(\cdot)$ is **Lambert W function**， ζ is the maximum degree of all k-hop CNs of $(i, j)$,the total number of paths of length ℓ between i and j is denoted as $η_l(i, j).$
> $$
> C =
> \frac{1}{\tbinom{\xi}{2}}
> \frac{D^{2k-1}}
>      {\eta_{2k} - D^{2k-2}
>         \dfrac{\sqrt{N\ln\delta^{-1}}}{4}},
> $$ and $D$ is the maximum degree on the graph.
>
> Plotting the curves in MATLAB shows that without introducing normalizedCN, the upper bound of $s_{ij}$ is a monotonically increasing affine function with respect to $k$. After introducing normalizedCN, this upper bound gradually decreases as $k$ increases. This extends our theoretical analysis to more realistic scenarios.
>
> ### W4.
> Thank you! We evaluated our model using the [n1] evaluation metric, and the results are as follows. It is noteworthy that [n1] essentially aggregates the outputs of many previous outstanding models. Nevertheless, our single model is able to compare competitively to [n1]'s ensemble model.
>
> |Method|Cora(HR@1)|Cora(HR@3)|Cora(HR@10)|Cora(HR@100)|
> |-|-|-|-|-|
> |Link-MoE|32.12±4.72|38.81±1.09|75.84±0.28|96.26±0.09|
> |OCN|35.10±5.22|49.44±4.86|65.91±2.08|89.82±0.91|
> |OCNP|39.21±5.71|50.75±6.07|65.19±1.32|90.06±1.01|
>
>
> |Method|Citeseer(HR@1)|Citeseer(HR@3)|Citeseer(HR@10)|Citeseer(HR@100)|
> |-|-|-|-|-|
> |Link-MoE|58.50±0.46|76.72±0.24|82.77±0.19|96.44±0.14|
> |OCN|57.70±0.22|78.65±0.98|86.11±0.79|93.62±1.30|
> |OCNP|60.79±0.70|74.53±0.42|87.40±1.14|93.41±1.02|
>
>
> |Method|Pubmed(HR@1)|Pubmed(HR@3)|Pubmed(HR@10)|Pubmed(HR@100)|
> |-|-|-|-|-|
> |Link-MoE|45.13±0.38|52.57±2.27|61.11±1.03|90.38±0.24|
> |OCN|62.52±2.19|69.23±0.60|72.74±1.69|83.96±0.51|
> |OCNP|48.25±1.26|68.02±0.99|72.08±2.32|82.32±1.21|
>
>
> |Method|Collab(HR@20)|Collab(HR@100)|PPA(HR@20)|PPA(HR@50)|
> |-|-|-|-|-|
> |Link-MoE|63.83±0.65|75.16±1.64|48.36±1.37|59.87±0.80|98.59±0.02  |99.28±0.01|99.63±0.02|
> |OCN|67.64±0.45|74.17±1.20|46.62±0.49|61.29±1.91|
> |OCNP|65.39±0.23|70.43±0.92|55.99±1.58|62.12±2.10|
>
> ### W5.
> Thanks! The results are as follows. As we can see, our models outperform [n2] on 5/6 datasets, and outperforms [n1] on 3/6 datasets.
>
>
> |Method|Cora(MRR)|Citeseer(MRR)|Pubmed(MRR)|Collab(HR@50)|PPA(HR@100)|Citation2(MRR)|
> |-|-|-|-|-|-|-|
> |LPFormer [n2]|39.42±5.78|65.42±4.65|40.17±1.92|68.14±0.51|63.32±0.63|89.81±0.13|
> |Link-MoE [n1]|44.03±2.28|67.49±0.30|53.10±0.24|71.32±0.99|69.39±0.61|91.25±0.02|
> |OCN|51.75±3.24|64.69±1.31|47.09±0.76|72.43±3.75|69.79±0.85|88.57±0.16|
> |OCNP|45.35±5.66|65.55±0.49|44.12±0.53|67.74±0.16|74.87±0.94|87.06±0.27|
>
> ### Q1.
> Thanks for your suggestion! The new results are as follows (hereinafter abbreviated as NC):
>
> |Method|Cora(HR@100)|Citeseer(HR@100)|Pubmed(HR@100)|Collab(HR@50) |PPA(HR@100)|Citation2(MRR)|DDI(HR@20)|
> |-|-|-|-|-|-|-|-|
> |CN|33.92±0.46|29.79±0.90|23.13±0.15|56.44|27.65|51.47|17.73|
> |AA|39.85±1.34|35.19±1.33|27.38±0.11|64.35|32.45|51.89|18.61|
> |RA|41.07±0.48|33.56±0.17|27.03±0.35|64.00|49.33|51.98|27.60|
> |NC|47.11|39.81|34.74|65.88|43.26|53.32|30.49|
>
> NC significantly outperforms the standard heuristics, demonstrating the importance of decorrelation and normalziation of high-order common neighbors.
>
> The superiority of NC over RA can also be demonstrated by the following example: Two hexagons are connected via a single node (Node 1). In the left hexagon, starting clockwise from the node adjacent to 1, the nodes are numbered consecutively as 2, 4, 6, 8, 10, 12. In the right hexagon, starting counterclockwise from the node adjacent to 1, the nodes are numbered consecutively as 3, 5, 7, 9, 11, 13. The computed values are: NC(13,5)=41/6,NC(13,1)=5/3,NC(13,9)=4/3.RA(13,5)=1/3,RA(13,1)=1/3,RA(13,9)=1/2. It is evident that the accuracy with which NC reflects the structural features is far better than RA.
>
> ### Q2.
> Thanks for raising this issue! The original expression was indeed inaccurate. The correct formulation should be: $\sum_{k=1}^K \alpha_k \{OCN^k \cdot \text{MPNN}(A, X)\}_{ij}.$
>
> Here, the preceding OCN is the $(m, n)$ matrix as previously defined, while $\text{MPNN}(A, X)$ is an $(n, \text{dim})$ matrix. The notation ${\cdot}_{ij}$ indicates the selection of the specific node pair $(i, j)$. We sincerely apologize for the oversight and greatly appreciate the identification of this error.
>
> ### Q3.
> The metrics adopted in our paper are widely accepted within the field. They follow the authoritative benchmark OGB (Open Graph Benchmark), which considers the different characteristics and purposes of the datasets and suggest different metrics. Our evaluation framework is also the officially recommended standard by OGB. These metrics have been utilized in nearly all influential works within the domain, such as SEAL, Neo-GNN, Buddy, NCN and [n2]. In contrast, a limited number of related works, such as [n1], employ their own evaluation metrics.

---

> > ### Comment · Reviewer_8Aox · 2025-08-04
> >
> > Dear authors,
> >
> > Thank you for your detailed response and for your efforts in addressing my comments, including the clarifications, the additional experimental comparisons with further methods, and the extension of your theoretical analysis to the Barabási–Albert graph model.
> >
> > Most of my concerns are now satisfactorily addressed, and I have therefore updated my evaluation to borderline accept. My stance is not fully positive, as I believe the clarity of the paper still requires improvement—a concern shared by several reviewers. I hope that, should the paper be accepted, the authors will take this feedback seriously and revise the manuscript accordingly.
> >
> > Best regards,
> >
> > Reviewer 8Aox

---

> > > ### Author Response · Authors · 2025-08-04
> > >
> > > Thank you very much for your recognition of our work! We are pleased that most of your concerns have been satisfactorily addressed. We will certainly treat all feedback with the utmost seriousness and revise the manuscript accordingly. Once again, we sincerely appreciate your valuable suggestions and wish you all the best in your work.

---

### Official Review · Reviewer_qjCp · 2025-07-04

**Clarity:** 2
**Significance:** 3
**Originality:** 3
**Rating:** 4
**Confidence:** 3

**Summary:**

This paper proposes a novel link prediction framework OCN, which leverages orthogonalized multi-hop common neighbor features. The authors identify and address the issues of redundancy and over-smoothing in higher-order common neighbors. By orthogonalizing the coefficients of different-order CNs, the model ensures a more expressive representation. Additionally, a path-based normalization technique is introduced to reduce the dominance of frequently appearing nodes. Experiments demonstrate competitive performance across multiple benchmark datasets.

**Questions:**

1. Both NCNC and SEAL already capture local structural information (NCNC through common neighbors and SEAL via enclosing subgraphs). Since OCN also uses K=2, could the authors clarify what specific relational patterns or advantages OCN captures that contribute to its superior performance over these baselines?

**Ethical Concerns:**

["NO or VERY MINOR ethics concerns only"]

**Limitations:**

yes

**Quality:**

3

**Strengths And Weaknesses:**

Strengths:

1. The idea of decorrelating multi-hop common neighbors via orthogonalization is reasonable and helps mitigate redundancy.

2. The proposed normalization strategy, inspired by batch normalization, is effective and empirically validated to reduce over-smoothing.

3. Empirical results across multiple benchmark datasets show competitive performance compared to recent baselines.


Weaknesses:

1. While the paper introduces mechanisms for handling high-order common neighbors, the final model only uses OCN¹ and OCN², limiting the demonstrated impact of the orthogonalization and normalization designs. This weakens the motivation for high-order modeling.

2. The explanation of key concepts such as “coefficient,” “orthogonalization,” and “path-based normalization” is not always directly clear. For instance, the coefficient definitions rely on abstract descriptions without concrete examples or diagrams. Based on the current description, one may infer that each node pair maintains a k × n coefficient matrix, but this is not explicitly stated.

3. The paper lacks clear explanation of how the values in Figure 1 and Figure 2 are obtained, which may hinder reader understanding of the motivation and insights.

---

> ### Author Rebuttal · Authors · 2025-07-31
>
> ### W1.
> Thank you very much for your question! The previously suboptimal performance of OCN-3 (shown in Table 2) stemmed from using the same hyperparameters as those for OCN, which failed to fully exploit the potential of higher-order CN. After further tuning the model using 3-hop CN with Optuna, we obtained some new good results. As shown in the below table, the tuned OCN-3 model indeed outperforms the original model on certain datasets. However, introducing higher-order CNs inevitably leads to a sharply increased computational burden and GPU memory consumption. Overall, we believe higher-order CNs can yield benefits, though this is also dependent on the specific model architecture and dataset characteristics. Therefore, we consider this work highly enlightening for subsequent research on how to efficiently utilize higher-order CNs.
>
>
> | Method      | Cora (HR@100) | Citeseer (HR@100) | Pubmed (HR@100) | Collab (HR@50) | PPA (HR@100) | Citation2 (MRR) | DDI (HR@20) |
> |-------------|---------------|-------------------|-----------------|----------------|--------------|-----------------|-------------|
> | OCN          | 89.82±0.91    | 93.62±1.30        | 83.96±0.51      | 72.43±3.75     | 69.79±0.85   | 88.57±0.06      | 97.42±0.34  |
> | OCN-3          | 90.96±0.64    | 92.33±0.85        | 87.12±1.40      | 73.04±2.13     | OOM   | 87.34±0.38      | 90.06±1.29  |
>
> ### W2.
> Thank you very much for your question and suggestions. The complete algorithm flow for orthogonalization and path-based normalization is presented in Algorithm 2 on page 34. The entire method relies on the definition of $CN^k$ (in line 156 and Appendix B). Specifically:
> $CN^k(u,v) = \sum_{2(k-1) < k_1 + k_2 \leq 2k, \atop k_1 \leq k, \ k_2 \leq k} \left(A^{k_1}\right)_u \odot \left(A^{k_2}\right)_v$
>
> The defined k-order CN is a scalar value, but it is defined for a specific node pair $(u, v)$. It represents: if a node has paths to $u$ of length $k₁$ and to $v$ of length $k₂$ (this refers to any path, not necessarily the shortest path), and the lengths $k₁$ and $k₂$ satisfy the given constraints, then this node is considered a k-order CN for the pair (u, v). The term $\left(A^{k_1}\right)_{u}$ (and similarly for $k_2$) actually counts the number of paths of length $k₁$ (or $k₂$) from this k-order CN node to $u$ (or $v$). This definition is also the commonly accepted definition of CN in the link prediction field, although some models define CN based on the Shortest Path Distance , whereas ours on the length of paths.
> What we actually orthogonalize is a matrix of shape $(h, n)$, where $h$ is the number of edges in the current batch, and n is the total number of nodes. In plain language, the definition in Appendix B means: For each order $k$, we have three matrices of shape $(h, n)$. e.g. for 4-hop CN, the distance pairs can be (4,4); (3,4); or (4,3). For 7-hop CN, (7,7); (6,7); or (7,6). For such an $(h, n)$ matrix, suppose it corresponds to the distance pair (3,4). Then, the value $z$ at row $x$ and column $y$ indicates that for the edge $(u, v)$ corresponding to row $x$, the total number of paths from the node corresponding to column $y$ to $u$ that are exactly of length 3, and to $v$ that are exactly of length 4, is $z$. If the total number of paths is 0, it means the node corresponding to column $y$ cannot reach $u$ and $v$ via any paths of lengths exactly (3,4) respectively. Consequently, this node is not a 4-hop CN under the (3,4) criterion. However, it might be a 4-hop CN under the (4,4) or (4,3) criteria. If it satisfies none of these criteria, then it is not a 4-hop CN for the pair $(u, v)$.
>
> Orthogonalization:
> The orthogonalization process is the standard Gram-Schmidt, but applied to matrices instead of vectors. The matrices we orthogonalize all have the shape $(m, n)$, where $m$ represents the total number of edges and $n$ represents the total number of nodes. Orthogonality between matrices is defined by the Frobenius norm:
> $$
> \|A^T B\|_F = 0 \quad \text{is equivalent to} \text{tr}(A^T B) = 0
> $$
> Section 4.1 primarily addresses one key challenge: how to obtain the true inner product value on a large real-world graph. Calculating the inner product for matrices used in orthogonalization inherently requires matrix multiplication. However, in real graphs, the number of edges $m$ and nodes $n$ is extremely large. Performing the full $(m, n)$ matrix multiplication across the entire graph is computationally expensive. Furthermore, training is performed batch-wise, so we only have access to a submatrix $(m', n)$ constructed from the edges within the current batch, where $m'$ is the number of edges in this batch.
> Our goal is therefore to approximate the inner product computed over all edges (i.e., the true inner product achieving orthogonality on the full graph) using only the partial information from the edges visible within the current batch. Inspired by Batch Normalization, we maintain a running inner product across batches. This running inner product accumulates the contribution of historical inner product information. It is then combined with the inner product calculated from the current limited information via a weighted sum. As the number of training steps $t$ tends towards infinity, this running inner product converges to the full-graph inner product. In our experiments, we observed that the inner products indeed converged to a stable value.
>
> 4.2 aims to find a method that achieves mutually orthogonal results—even if approximately—while bypassing the Gram-Schmidt. Because Gram-Schmidt requires each subsequent matrix to operate on every preceding orthogonal basis, we desire a way to directly compute the orthogonalized result from each $(m, n)$ matrix in a single step. Through proof, we discovered that simply applying $OCN^k = CN^k \cdot \text{diag}(T_k)$—that is, right-multiplying the $(m, n)$ matrix by a diagonal matrix (with an arbitrarily chosen polynomial basis placed on its diagonal)—sufficiently approximates the result originally requiring Gram-Schmidt. This reduces the time complexity from $O(k^2)$ to $O(k)$ (see Table 7).
>
> Path-based Normalization:
> The exact steps and equations used for normalization in Section 5 are detailed in Definition 5.1. We provide accessible descriptions of the underlying concepts and their practical implications in lines 212-221. The precise, step-by-step computational workflow for path-based normalization is presented within the first for loop of Algorithm 2 on page 34.
> Thank you very much once again for your questions and suggestions!
>
> ### W3.
> Thank you very much for question. The data in Fig.1 come from Cora. The correlation coefficients used are Pearson correlation coefficients. The data in Fig.2 are also sourced from Cora. We also computed the relevant metrics on other datasets and found that they exhibit similar patterns. Therefore, we only showed Cora.
> The purpose of Fig.2 is to illustrate the difference between using path-based normalization and not using it. The coefficient of variation (CV) decreases at higher orders, indicating that the high-order CNs of different nodes begin to overlap more frequently and that the similarity of higher-order neighborhood structures continually increases. Our path-based normalization method can significantly slow this decreasing trend of the CV.
>
> ### Q1.
> Thank you for your question!
> As for empirical advantages of OCN, they primarily come from the following points. Firstly, OCN captures the feature information of the high-order CNs, rather than only their counts. Secondly, the information coming from CNs have undergone decorrelation, thus increasing their utilization by models. Thirdly, the weight (significance) assigned to each high-order CN is dynamically adjusted (If $k$-hop CNs are less frequently shared among other node pairs, then these $k$-hop CNs carry greater discriminative significance for the relationship between the two nodes.).
> Previous work generally treated high-order CNs only as a source of information of count, failing to fully exploit the rich information inherent in the CNs. Furthermore, information across different high-order CNs often contains substantial redundancy and coupling. Our model preserves the most effective information while performing decorrelation. Simultaneously, our model does not assign equal weight to every CN; we recognize that the importance of each CN varies. This approach also helps mitigate the mentioned over-smoothing phenomenon. The redundancy and over-smoothing phenomena are two intriguing issues we identified; addressing them is a key reason our model outperforms other baselines. The above points are all verified by ablation study in Table 2 and main results in Table 1.
> We hope this answer addresses your query. Further discussion is welcome!

---

> > ### Comment · Reviewer_PbQ7 · 2025-08-09
> >
> > We appreciate the clarifications. The proposed method is definitely interesting and useful. Looking forward to publication of your work.

---

> > > ### Author Response · Authors · 2025-08-09
> > >
> > > Dear Reviewer PbQ7,
> > >
> > > Thank you very much for your attention to our work. We noticed that due to your busy schedule, you may have forgotten to update your previous score.  We greatly appreciate the time and effort you have devoted!
> > >
> > > Best regards,
> > >
> > > Submission 6119 Authors

---

> ### Comment · Reviewer_qjCp · 2025-08-04
> **Official Comment by Reviewer qjCp**
>
> Dear authors,
>
> Thank you for the detailed response and for your efforts in addressing my comments. Most of my concerns have been adequately addressed. While the introduction of OCN-3 aims to explore the role of higher-order CN, the empirical results for order 3 are not particularly strong, suggesting that the practical benefits of this extension remain to be further validated. As also noted by several other reviewers, the clarity and presentation of the paper still require further improvement. I will therefore maintain my current ratings.
>
> Best regards,
>
> Reviewer qjCp.

---

> > ### Author Response · Authors · 2025-08-05
> >
> > Thank you for your recognition and feedback! Due to space constraints, we have moved some precise expressions, algorithms, and detailed explanations to the Appendix. In the revised version, we will make certain parts of the main text clearer and more accessible, while reiterating some key details from the Appendix.
> > Thank you again, and wish you all the best!

---

### Author Response · Authors · 2025-08-07

Dear Reviewers,

We sincerely appreciate the insightful comments and constructive feedback from all of you. We hope that our revisions and clarifications have adequately responded to your concerns.

Key Contributions of Our Original Work:

As summarized by Reviewer:
> our paper proposes "**nice and simple heuristics** to improve performance of link prediction", "**two effective approaches** which reduces redundancy and homogeneity", and recognizing that "the proposed method definitely is **useful and effective**", "the paper has **extensive experimental results** to demonstrate the effectiveness"

The most important precondition is, **previous works exhibited low utilization efficiency and suboptimal effectiveness** when incorporating higher-order CNs, sometimes even hurting the performance. Our focus is thus **Effectively Utilizing Higher-Order CNs**. We systematically investigated why existing methods for incorporating higher-order CNs underperform. Our analysis revealed two key phenomena central to our study: redundancy and over-smoothing. Accordingly, we propose two techniques: scalable orthogonalization and path-based normalization, both with **solid mathematical and theoretical foundations** to justify their effectiveness, and are **nontrivially adapted to large graphs** for scalability. Our model achieves both high performance through effective utilization of high-order CNs as well as high efficiency!

During the rebuttal period, we were also pleased to learn that **most concerns have been adequately addressed through our responses and additional experiments**. Thank you very much!

Regarding some raised questions about the clarity of the paper, answers to almost all doubts about specific concepts or definitions in the article can actually be found in our original text (possibly in the appendix). Due to the page limit for submissions, we had to **place many explanations and detailed discussions in the appendix**, retaining only concise formulas and succinct yet essential statements in the main text. We greatly appreciate your constructive and valuable suggestions! Sincere thanks to all for your attention to this work!

Best regards,

Submission 6119 Authors

---

### Decision · Program_Chairs · 2025-09-17

**Decision:**

Accept (poster)

**Comment:**

The work offers useful, principled improvements for a widely used structural signal in link prediction. The method is simple, well-motivated, and effective, with ablations and theory. The remaining issues post-rebuttal are presentation-centric rather than technical: with clearer definitions, worked examples, and a compact cost and metrics summary, the contribution will be accessible and reproducible.

For the next version, I’d suggest: 1) making the core objects concrete with a toy graph, 2) adding a short subsection on the batch inner-product estimator such as what you store, the update rule, a one-line equation, and when it matches full-graph orthogonality in practice, 3) unifying reporting with two compact tables: one mapping each dataset to its metric, and one cost table with parameter count, training time, and peak GPU memory. In addition, you can consider making cosmetic chanes like tightening writing and figures and sharpening related-work positioning.